# A Simple Video Segmenter by Tracking Objects Along Axial Trajectories

**Ju He**                                                                    *jhe47@jhu.edu*
*Johns Hopkins University*

**Qihang Yu**                                                    *qihang.yu@bytedance.com*
*ByteDance*

**Inkyu Shin**                                                      *dlsrbgg33@kaist.ac.kr*
*Korea Advanced Institute of Science and Technology*

**Xueqing Deng**                                              *xueqingdeng@bytedance.com*
*ByteDance*

**Alan Yuille**                                                              *ayuille1@jhu.edu*
*Johns Hopkins University*

**Xiaohui Shen**                                              *shenxiaohui@bytedance.com*
*ByteDance*

**Liang-Chieh Chen**                                    *liangchieh.chen@bytedance.com*
*ByteDance*

**Reviewed on OpenReview:** *https://openreview.net/forum?id=Sy6ZOStz5v*

## Abstract

Video segmentation requires consistently segmenting and tracking objects over time. Due to the quadratic dependency on input size, directly applying self-attention to video segmentation with high-resolution input features poses significant challenges, often leading to insufficient GPU memory capacity. Consequently, modern video segmenters either extend an image segmenter without incorporating any temporal attention or resort to window space-time attention in a naive manner. In this work, we present Axial-VS, a general and simple framework that enhances video segmenters by tracking objects along axial trajectories. The framework tackles video segmentation through two sub-tasks: short-term within-clip segmentation and long-term cross-clip tracking. In the first step, Axial-VS augments an off-the-shelf clip-level video segmenter with the proposed *axial-trajectory* attention, sequentially tracking objects along the height- and width-trajectories within a clip, thereby enhancing temporal consistency by capturing motion trajectories. The axial decomposition significantly reduces the computational complexity for dense features, and outperforms the window space-time attention in segmentation quality. In the second step, we further employ *axial-trajectory* attention to the object queries in clip-level segmenters, which are learned to encode object information, thereby aiding object tracking across different clips and achieving consistent segmentation throughout the video. Without bells and whistles, Axial-VS showcases state-of-the-art results on video segmentation benchmarks, emphasizing its effectiveness in addressing the limitations of modern clip-level video segmenters. Code and models are available here.

# 1   Introduction

Video segmentation is a challenging computer vision task that requires temporally consistent pixel-level scene understanding by segmenting objects, and tracking them across a video. Numerous approaches have been proposed to address the task in a variety of ways. They can be categorized into frame-level (Kim et al., 2020; Wu et al., 2022c; Heo et al., 2023; Li et al., 2023a), clip-level (Athar et al., 2020; Qiao et al., 2021; Hwang et al., 2021; Mei et al., 2022), and video-level segmenters (Wang et al., 2021b; Heo et al., 2022; Zhang et al., 2023), which process the video either in a frame-by-frame, clip-by-clip, or whole-video manner.

Among them, clip-level segmenters draw our special interest, as it innately captures the local motion within a short period of time (a few frames in the same clip) compared to frame-level segmenters. It also avoids the memory constraints incurred by the video-level segmenters when processing long videos. Specifically, clip-level segmenters first pre-process the video into a set of short clips, each consisting of just a few frames. They then predict clip-level segmentation masks and associate them (*i.e.*, tracking objects across clips) to form the final temporally consistent video-level results.

Concretely, the workflow of clip-level segmenters requires two types of tracking: short-term *within-clip* and long-term *cross-clip* tracking. Most existing clip-level segmenters (Li et al., 2023b; Shin et al., 2024) directly extend the modern image segmentation models (Cheng et al., 2022; Yu et al., 2022b) to clip-level segmentation without any temporal attention, while TarViS (Athar et al., 2023) leverages a straightforward window space-time attention mechanism for within-clip tracking. However, none of the previous studies have fully considered the potential to enhance within-clip tracking and ensure long-term consistent tracking beyond neighboring clips. An intuitive approach to improve tracking ability is to naively calculate the affinity between features of neighboring frames (Vaswani et al., 2017). Another unexplored direction involves tracking objects along trajectories, where a variant of self-attention called trajectory attention (Patrick et al., 2021) was proposed to capture object motion by computing the affinity of down-sampled embedded patches in video classification. Nevertheless, in video segmentation, the input video is typically of high-resolution and considerable length. Due to the quadratic complexity of attention computation concerning input size, directly computing self-attention or trajectory attention for dense pixel features becomes computationally impractical.

To address this challenge, we demonstrate the feasibility of decomposing and detecting object motions independently along the height (H-axis) and width (W-axis) dimensions, as illustrated in Fig. 1. This approach sequentially computes the affinity between features of neighboring frames along the height and width dimensions, a concept we refer to as *axial-trajectory* attention. The axial-trajectory attention is designed to learn the temporal correspondences between neighboring frames by estimating the motion paths sequentially along the height- and width-axes. By concurrently considering spatial and temporal information in the video, this approach harnesses the potential of attention mechanisms for dense pixel-wise tracking. Furthermore, the utilization of axial-trajectory attention can be expanded to compute the affinity between clip object queries. Modern clip-level segmenters encode object information in clip object queries, making this extension valuable for establishing robust long-term cross-clip tracking. These innovations serve as the foundations for our within-clip and cross-clip tracking modules. Building upon these components, we introduce Axial-VS, a general and simple framework for video segmentation. Axial-VS enhances a clip-level segmenter by incorporating within-clip and cross-clip tracking modules, leading to exceptional temporally consistent segmentation results. This comprehensive approach showcases the efficacy of axial-trajectory attention in addressing both short-term within-clip and long-term cross-clip tracking requirements.

We instantiate Axial-VS by employing Video-kMaX (Shin et al., 2024) or Tube-Link (Li et al., 2023b) as the clip-level segmenters, yielding a significant improvement on video panoptic segmentation (Kim et al., 2020) and video instance segmentation (Yang et al., 2019), respectively. Without bells and whistles, Axial-VS improves over Video-kMaX (Shin et al., 2024) by 8.5% and 5.2% VPQ on VIPSeg (Miao et al., 2022) with ResNet50 (He et al., 2016) and ConvNeXt-L (Liu et al., 2022b), respectively. Moreover, it also achieves 3.5% VPQ improvement on VIPSeg compared to the state-of-the-art model DVIS (Zhang et al., 2023), when using ResNet50. Besides, Axial-VS can also boost the strong baseline Tube-Link (Li et al., 2023b) by 0.9% AP, 4.7% AP$^{\text{long}}$, and 6.5% AP on Youtube-VIS-2021 (Yang et al., 2021a), Youtube-VIS-2022 (Yang et al., 2022), and OVIS (Qi et al., 2022) with Swin-L (Liu et al., 2021).

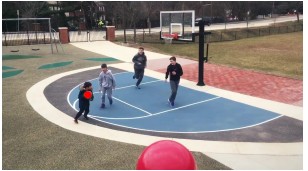 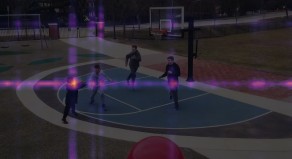 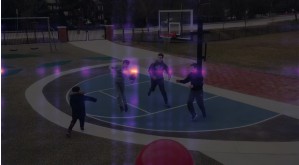 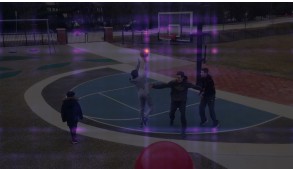

a) Selected *reference point* at frame 1    b) *axial-trajectory attention* at frame 2    c) *axial-trajectory attention* at frame 3    d) *axial-trajectory attention* at frame 4

Figure 1: **Visualization of Learned Axial-Trajectory Attention.** In this short clip depicting the action 'playing basketball', the basketball location at frame 1 is selected as the *reference point* (mark in red). We multiply the learned height and width axial-trajectory attentions and overlay them on frame 2, 3 and 4 to visualize the trajectory of the *reference point* over time. As observed, the axial-trajectory attention can capture the basketball's motion path.

## 2 Related Work

**Attention for Video Classification**  The self-attention mechanism (Vaswani et al., 2017) is widely explored in the modern video transformer design (Bertasius et al., 2021; Arnab et al., 2021; Neimark et al., 2021; Fan et al., 2021; Patrick et al., 2021; Liu et al., 2022a; Wang & Torresani, 2022) to reason about the temporal information for video classification. While most works treat time as just another dimension and directly apply global space-time attention, the divided space-time attention (Bertasius et al., 2021) applies temporal attention and spatial attention separately to reduce the computational complexity compared to the standard global space-time attention. Trajectory attention (Patrick et al., 2021) learns to capture the motion path of each query along the time dimension. Deformable video transformer (Wang & Torresani, 2022) exploits the motion displacements encoded in the video codecs to guide where each query should attend in their deformable space-time attention. However, most of the aforementioned explorations cannot be straightforwardly extended to video segmentation due to the quadratic computational complexity and the high-resolution input size of videos intended for segmentation. In this study, we innovatively suggest decomposing object motion into height and width-axes separately, thereby incorporating the concept of axial-attention (Ho et al., 2019; Huang et al., 2019; Wang et al., 2020). This leads to the proposal of axial-trajectory attention, effectively enhancing temporal consistency while maintaining manageable computational costs.

**Attention for Video Segmentation**  The investigation into attention mechanisms for video segmentation is under-explored, primarily hindered by the high-resolution input size of videos. Consequently, most existing works directly utilizes the modern image segmentation models (Cheng et al., 2022) to produce frame-level or clip-level predictions, while associating the cross-frame or cross-clip results through Hungarian Matching (Kuhn, 1955). VITA (Heo et al., 2022) utilizes window-based self-attention to effectively capture relations between cross-frame object queries in the object encoder. DVIS (Zhang et al., 2023) similarly investigates the use of standard self-attention to compute the affinity between cross-frame object queries, resulting in improved associating outcomes. TarViS (Athar et al., 2023) introduces a window space-time attention mechanism at the within-clip stage. Axial-VS extends these concepts by introducing the computation of axial-trajectory attention along object motion trajectories sequentially along the height and width axes. This operation is considered more effective for improving within-clip tracking capabilities and facilitating simultaneous reasoning about temporal and spatial relations. Moreover, we apply axial-trajectory attention to object queries to efficiently correlate cross-clip predictions, thereby enhancing cross-clip consistency.

**Video Segmentation**  Video segmentation aims to achieve consistent pixel-level scene understanding throughout a video. The majority of studies in this field primarily focus on video instance segmentation, addressing the challenges posed by 'thing' instances. Additionally, video panoptic segmentation is also crucial, emphasizing a comprehensive understanding that includes both 'thing' and 'stuff' classes. Both video instance and panoptic segmentation employ similar tracking modules, and thus we briefly introduce them together. Based on the input manner, they can be roughly categorized into frame-level segmenters (Yang et al., 2019; Kim et al., 2020; Yang et al., 2021b; Ke et al., 2021; Fu et al., 2021; Li et al., 2022; Wu et al., 2022c; Huang et al., 2022; Heo et al., 2023; Liu et al., 2023; Ying et al., 2023; Li et al., 2023a), clip-level segmenters (Athar et al., 2020; Qiao et al., 2021; Hwang et al., 2021; Wu et al., 2022a; Mei et al., 2022; Athar et al., 2023;

Li et al., 2023b; Shin et al., 2024), and video-level segmenters (Wang et al., 2021b; Lin et al., 2021; Wu et al., 2022b; Heo et al., 2022; Zhang et al., 2023). Specifically, TubeFormer (Kim et al., 2022) tackles multiple video segmentation tasks in a unified manner (Wang et al., 2021a), while TarVIS (Athar et al., 2023) proposes task-independent queries. Tube-Link (Li et al., 2023b) exploits contrastive learning to better align the cross-clip predictions. Video-kMaX (Shin et al., 2024) extends the image segmenter (Yu et al., 2022b) for clip-level video segmentation. VITA (Heo et al., 2022) exhibits a video-level segmenter framework by introducing a set of video queries. DVIS (Zhang et al., 2023) proposes a referring tracker to denoise the frame-level predictions and a temporal refiner to reason about long-term tracking relations. Our work focuses specifically on improving clip-level segmenters, and is thus mostly related to the clip-level segmenters Video-kMaX (Shin et al., 2024) and Tube-Link (Li et al., 2023b). Building on top of them, Axial-VS proposes the within-clip and cross-clip tracking modules for enhancing the temporal consistency within each clip and over the whole video, respectively. Our cross-clip tracking module is similar to VITA (Heo et al., 2022) and DVIS (Zhang et al., 2023) in the sense that object queries are refined to obtain the final video outputs. However, our model builds on top of clip-level segmenters instead of frame-level segmenters, and we use axial-trajectory attention to refine the object queries without extra complex designs, while VITA introduces another set of video queries and DVIS additionally cross-attends to the queries cashed in the memory.

# 3 Method

In this section, we briefly overview the clip-level video segmenter framework in Sec. 3.1. We then introduce the proposed *within-clip* tracking and *cross-clip* tracking modules in Sec. 3.2 and Sec. 3.3, respectively.

## 3.1 Video Segmentation with Clip-level Segmenter

**Formulation of Video Segmentation** Recent works (Kim et al., 2022; Li et al., 2022) have unified different video segmentation tasks as a simple set prediction task (Carion et al., 2020), where the input video is segmented into a set of tubes (a tube is obtained by linking segmentation masks along the time axis) to match the ground-truth tubes. Concretely, given an input video $\mathbf{V} \in \mathbb{R}^{L \times 3 \times H \times W}$ with $L$ represents the video length and $H$, $W$ represent the frame height and width, video segmentation aims at segmenting it into a set of $N$ class-labeled tubes:

$$\{\hat{y}_i\} = \{(\hat{m}_i, \hat{p}_i(c))\}_{i=1}^N, \tag{1}$$

where $\hat{m}_i \in [0,1]^{L \times H \times W}$ and $\hat{p}_i(c)$ represent the predicted tube and its corresponding semantic class probability. The ground-truth set containing $M$ class-labeled tubes is similarly represented as $\{y_i\} = \{(m_i, p_i(c))\}_{i=1}^M$. These two sets are matched through Hungarian Matching (Kuhn, 1955) during training to compute the losses.

**Formulation of Clip-Level Video Segmentation** The above video segmentation formulation is theoretically applicable to any length $L$ of video sequences. However, in practice, it is infeasible to fit the whole video into modern large network backbones during training. As a result, most works exploit frame-level segmenter or clip-level segmenter (a clip is a short video sequence typically of two or three frames) to get frame-level or clip-level tubes first and further associate them to obtain the final video-level tubes. In this work, we focus on the clip-level segmenter, since it better captures local temporal information between frames in the same clip. Formally, we split the whole video $\mathbf{V}$ into a set of *non-overlapping* clips: $v_i \in \mathbb{R}^{T \times 3 \times H \times W}$, where $T$ represents the length of each clip in temporal dimension (assuming that $L$ is divisible by $T$ for simplicity; if not, we simply duplicate the last frame). For the clip-level segmenter, we require $T \geq 2$.

**Overview of Proposed Axial-VS** Given the independently predicted clip-level segmentation, we propose Axial-VS, a meta-architecture that builds on top of an off-the-shelf clip-level segmenter (*e.g.*, Video-kMaX (Shin et al., 2024) or Tube-Link (Li et al., 2023b)) to generate the final temporally consistent video-level segmentation results. Building on top of the clip-level segmenter, Axial-VS contains two additional modules: within-clip tracking module and cross-clip tracking module, as shown in Fig. 2. We detail each module in the following subsections, and choose Video-kMaX as the baseline for simplicity in describing the detailed designs.

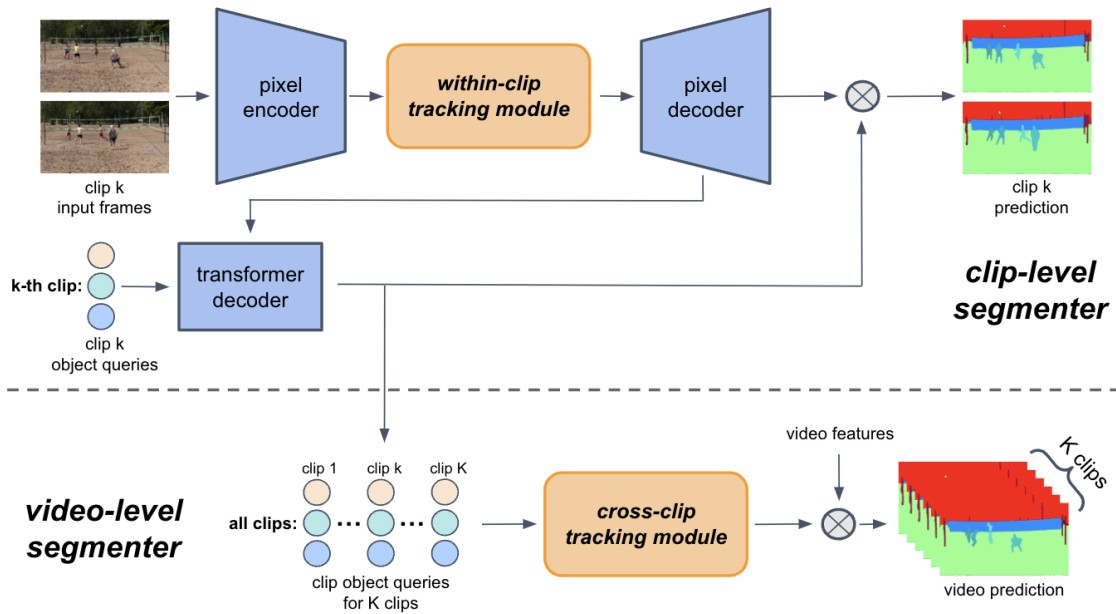

Figure 2: **Overview of Axial-VS,** which builds two components on top of a clip-level segmenter (blue): the *within-clip tracking* and *cross-clip tracking modules* (orange). Both modules exploit the axial-trajectory attention to enhance temporal consistency. We obtain video features by concatenating all clip features output by the pixel decoder (totally $K$ clips), and video prediction by multiplying ($\otimes$) video features and refined clip object queries.

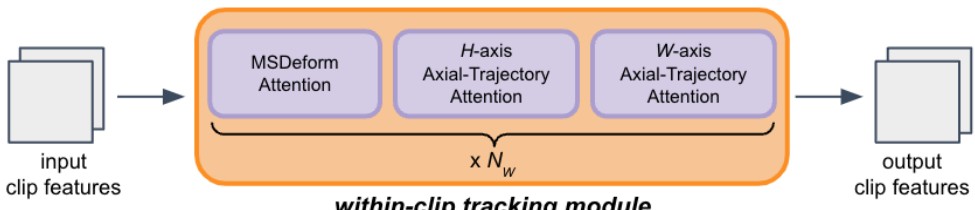

Figure 3: **Within-clip tracking module** takes input clip features extracted by the network backbone, iteratively stacks Multi-Scale Deformable (MSDeform) Attention and axial-trajectory attention (sequentially along H- and W-axes) for $N_w$ times, and outputs the spatially and temporally consistent clip features.

## 3.2 Within-Clip Tracking Module

As shown in Fig. 3, the main component of the within-clip tracking module is the proposed *axial-trajectory attention*, which decomposes the object motion in the height-axis and width-axis, and effectively learns to track objects across the frames in the same clip (thus called *within-clip* tracking). In the module, we also enrich the features by exploiting the multi-scale deformable attention (Zhu et al., 2020) to enhance the spatial information extraction. We explain the module in detail below.

**Axial-Trajectory Attention**   Trajectory attention (Patrick et al., 2021) was originally proposed to capture the object motion information contained in the video for the classification task. However, unlike video classification, where input video is usually pre-processed into a small set of tokens and the output prediction is a single label, video segmentation requires dense prediction (*i.e.*, per pixel) results, making it infeasible to directly apply trajectory attention, which has quadratic complexity proportional to the input size. To unleash the potential of tracking objects through attention in video segmentation, we propose *axial-trajectory attention* that tracks objects along axial trajectories, which not only effectively captures object motion information but also reduces the computational cost.

Formally, given an input video clip consisting of $T$ frames, we forward it through a frame-level network backbone (*e.g.*, ConvNeXt (Liu et al., 2022b)) to extract the feature map $\mathbf{F} \in \mathbb{R}^{T \times D \times H \times W}$, where $D, H, W$ stand for the dimension, height and width of the feature map, respectively. We note that the feature map $\mathbf{F}$ is extracted frame-by-frame via the network backbone, and thus no temporal information exchanges between frames. We further reshape the feature into $\mathbf{F}_h \in \mathbb{R}^{W \times TH \times D}$ to obtain a sequence of $TH$ pixel features $\mathbf{x}_{th} \in \mathbb{R}^D$. Following (Vaswani et al., 2017), we linearly project $\mathbf{x}_{th}$ to a set of query-key-value vectors $\mathbf{q}_{th}, \mathbf{k}_{th}, \mathbf{v}_{th} \in \mathbb{R}^D$. We then perform *axial-attention* along trajectories (*i.e.*, the probabilistic path of a point between frames). Specifically, we choose a specific time-height position $th$ as the *reference point* to illustrate the computation process of axial-trajectory attention. After obtaining its corresponding query $\mathbf{q}_{th}$, we construct a set of trajectory points $\widetilde{y}_{tt'h}$ which represents the pooled information weighted by the trajectory probability. The *axial-trajectory* extends for the duration of the clip, and its point $\widetilde{y}_{tt'h} \in \mathbb{R}^D$ at different times $t'$ is defined as:

$$\widetilde{\mathbf{y}}_{tt'h} = \sum_{h'} \mathbf{v}_{t'h'} \cdot \frac{\exp \langle \mathbf{q}_{th}, \mathbf{k}_{t'h'} \rangle}{\sum_{\bar{h}} \exp \langle \mathbf{q}_{th}, \mathbf{k}_{t'\bar{h}} \rangle}. \tag{2}$$

Note that this step computes the axial-trajectory attention in $H$-axis (index $h'$), independently for each frame. It finds the axial-trajectory path of the reference point $th$ across frames $t'$ in the clip by comparing the reference point's query $\mathbf{q}_{th}$ to the keys $\mathbf{k}_{t'h'}$, only along the $H$-axis.

To reason about the intra-clip connections, we further pool the trajectories over time $t'$. Specifically, we linearly project the trajectory points and obtain a new set of query-key-value vectors:

$$\widetilde{\mathbf{q}}_{th} = \mathbf{W}_q \widetilde{\mathbf{y}}_{tth}, \quad \widetilde{\mathbf{k}}_{tt'h} = \mathbf{W}_k \widetilde{\mathbf{y}}_{tt'h}, \quad \widetilde{\mathbf{v}}_{tt'h} = \mathbf{W}_v \widetilde{\mathbf{y}}_{tt'h}, \tag{3}$$

where $\mathbf{W}_q, \mathbf{W}_k$, and $\mathbf{W}_v$ are the linear projection matrices for query, key, and value. We then update the reference point at time-height $th$ position by applying 1D attention along the time $t'$:

$$\mathbf{y}_{th} = \sum_{t'} \widetilde{\mathbf{v}}_{tt'h} \cdot \frac{\exp \langle \widetilde{\mathbf{q}}_{th}, \widetilde{\mathbf{k}}_{tt'h} \rangle}{\sum_{\bar{t}} \exp \langle \widetilde{\mathbf{q}}_{th}, \widetilde{\mathbf{k}}_{t\bar{t}h} \rangle}. \tag{4}$$

With the above update rules, we propagate the motion information in $H$-axis in the video clip. To capture global information, we further reshape the feature into $\mathbf{F}_w \in \mathbb{R}^{H \times TW \times D}$ and apply the same axial-trajectory attention (but along the $W$-axis) consecutively to capture the width dynamics as well.

The proposed axial-trajectory attention (illustrated in Fig. 4) effectively reduces the computational complexity of original trajectory attention from $\mathcal{O}(T^2 H^2 W^2)$ to $\mathcal{O}(T^2 H^2 W + T^2 W^2 H)$, allowing us to apply it to the dense video feature maps, and to reason about the motion information across frames in the same clip.

**Multi-Scale Attention**   To enhance the features spatially, we further adopt the multi-scale deformable attention (Zhu et al., 2020) for exchanging information at different scales of feature. Specifically, we apply the multi-scale deformable attention to the feature map $\mathbf{F}$ (extracted by the network backbone) frame-by-frame,

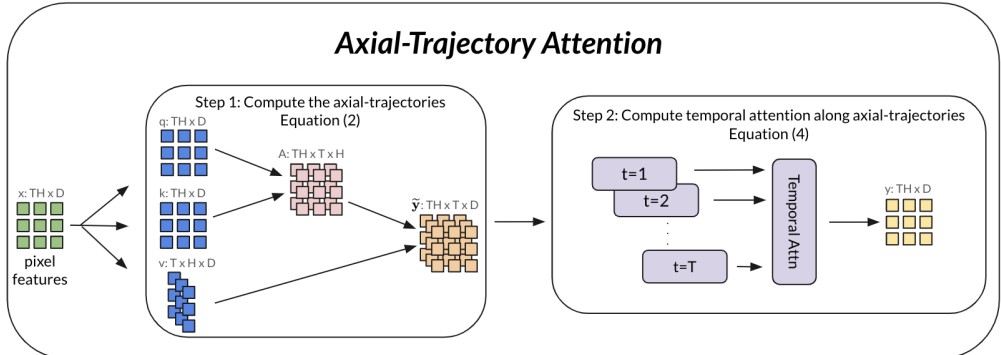

Figure 4: **Illustration of Axial-Trajectory Attention** (only *Height*-axis axial-trajectory attention is shown for simplicity), which includes two steps: computing the axial-trajectories $\widetilde{y}$ along *Height*-axis (Eq. 2) of the dense pixel feature maps $x \in \mathbb{R}^{TH \times D}$, where $T$, $H$, and $D$ denote the clip length, feature height and channels, respectively and then computing temporal attention along the axial-trajectories (Eq. 4) to obtain the temporally consistent features $y$.

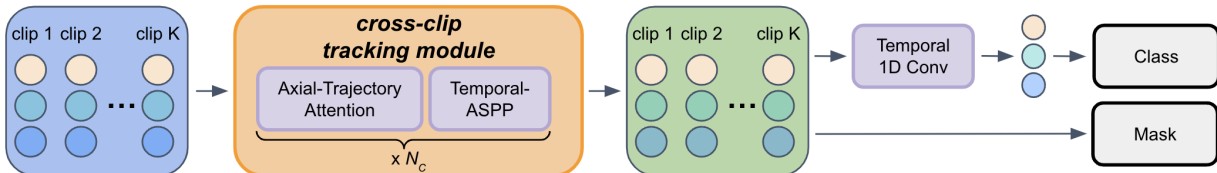

Figure 5: **Cross-clip tracking module** refines K sets of clip object queries by performing axial-trajectory attention and temporal atrous spatial pyramid pooing (Temporal-ASPP) for $N_c$ times.

which effectively exchanges the information across feature map scales (stride 32, 16, and 8) for each frame. In the end, the proposed within-clip tracking module is obtained by iteratively stacking multi-scale deformable attention and axial-trajectory attention (for $N_w$ times) to ensure that the learned features are spatially consistent across the scales and temporally consistent across the frames in the same clip.

**Transformer Decoder**  After extracting the spatially and temporally enhanced features, we follow typical video mask transformers (*e.g.*, Video-kMaX (Shin et al., 2024) or Tube-Link (Li et al., 2023b)) to produce clip-level predictions, where clip object queries $\mathbf{C}_k \in \mathbb{R}^{N \times D}$ (for $k$-th clip) are iteratively refined by multiple transformer decoder layers (Carion et al., 2020). The resulting *clip object queries* are used to generate a set of $N$ class-labeled tubes within the clip, as described in Sec. 3.1.

**Clip-Level (Near-Online) Inference**  With the above within-clip tracking module, our clip-level segmenter is capable of segmenting the video in a near-online fashion (*i.e.*, clip-by-clip). Unlike Video-kMaX (Shin et al., 2024) which takes overlapping clips as input and uses video stitching (Qiao et al., 2021) to link predicted clip-level tubes, our method simply uses the Hungarian Matching (Kuhn, 1955) to associate the clip-level tubes via the clip object queries (similar to MinVIS (Huang et al., 2022); but we work on the clip-level, instead of frame-level), since our input clips are non-overlapping.

## 3.3 Cross-Clip Tracking Module

Though axial-trajectory attention along with the multi-scale deformable attention effectively improves the within-clip tracking ability, the inconsistency between clips (*i.e.*, beyond the clip length $T$) still remains a challenging problem, especially under the fast-moving or occluded scenes. To address these issues, we further propose a cross-clip tracking module to refine and better associate the clip-level predictions. Concretely, given all the clip object queries $\{\mathbf{C}_k\}_{k=1}^K \in \mathbb{R}^{KN \times D}$ of a video (which is divided into $K = L/T$ non-overlapping clips, and $k$-th clip has its own clip object queries $\mathbf{C}_k \in \mathbb{R}^{N \times D}$), we first use the Hungarian Matching to align the clip object queries as the initial tracking results (*i.e.*, "clip-level inference" in Sec. 3.2). Subsequently, these results are refined by our proposed cross-clip tracking module to capture temporal connections across

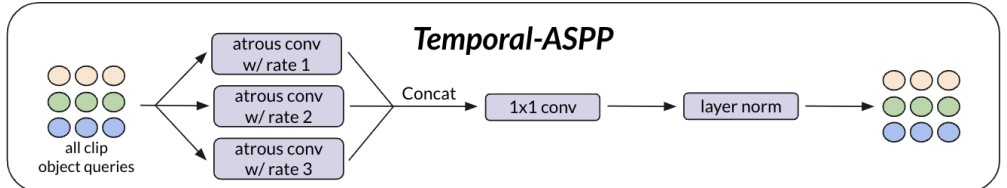

Figure 6: **Illustration of Temporal-ASPP,** which operates on the clip object queries and includes three parallel atrous convolution with different atrous rates to aggregate local temporal cross-clip connections across different time spans followed by 1x1 convolution and layer norm to obtain the final updated clip object queries.

the entire video, traversing all clips. As shown in Fig. 5, the proposed cross-clip tracking module contains two operations: axial-trajectory attention and Temporal Atrous Spatial Pyramid Pooling (Temporal-ASPP). We elaborate on each operation in detail below.

**Axial-Trajectory Attention** For $k$-th clip, the clip object queries $\mathbf{C}_k$ encode the clip-level tube predictions (*i.e.*, each query in $\mathbf{C}_k$ generates the class-labeled tube for a certain object in $k$-th clip). Therefore, associating clip-level prediction results is similar to finding the trajectory path of object queries in the whole video. Motivated by this observation, we suggest leveraging axial-trajectory attention to capture whole-video temporal connections between clips. This can be accomplished by organizing all clip object queries in a sequence based on the temporal order (*i.e.*, clip index) and applying axial-trajectory attention along the sequence to infer global cross-clip connections. Formally, for a video divided into $K$ clips (and each clip is processed by $N$ object queries), each object query $\mathbf{C}_{kn} \in \{\mathbf{C}_k\}$ is first projected into a set of query-key-value vectors $\mathbf{q}_{kn}, \mathbf{k}_{kn}, \mathbf{v}_{kn} \in \mathbb{R}^D$. Then we compute a set of trajectory queries $\widetilde{\mathbf{Z}}_{kk'n}$ by calculating the probabilistic path of each object query:

$$\widetilde{\mathbf{Z}}_{kk'n} = \sum_{k'} \mathbf{v}_{k'n'} \cdot \frac{\exp \langle \mathbf{q}_{kn}, \mathbf{k}_{k'n'} \rangle}{\sum_{\overline{n}} \exp \langle \mathbf{q}_{kn}, \mathbf{k}_{k'\overline{n}} \rangle}. \tag{5}$$

After further projecting the trajectory queries $\widetilde{Z}_{kk'n}$ into $\widetilde{\mathbf{q}}_{kn}, \widetilde{\mathbf{k}}_{kk'n}, \widetilde{\mathbf{v}}_{kk'n}$, we aggregate the cross-clip connections along the trajectory path of object queries through:

$$Z_{kn} = \sum_{k'} \widetilde{\mathbf{v}}_{kk'n} \cdot \frac{\exp \langle \widetilde{\mathbf{q}}_{kn}, \widetilde{\mathbf{k}}_{kk'n} \rangle}{\sum_{\overline{k}} \exp \langle \widetilde{\mathbf{q}}_{kn}, \widetilde{\mathbf{k}}_{k\overline{k}n} \rangle}. \tag{6}$$

**Temporal-ASPP** While the above axial-trajectory attention reasons about the whole-video temporal connections, it can be further enriched by a short-term tracking module. Motivated by the success of the atrous spatial pyramid pooling (ASPP (Chen et al., 2017a; 2018)) in capturing spatially multi-scale context information, we extend it to the temporal domain. Specifically, our Temporal-ASPP module contains three parallel temporal atrous convolutions (Chen et al., 2015; 2017b) with different rates applied to the all clip object queries $\mathbf{Z}$ for capturing motion at different time spans, as illustrated in Fig. 6.

**Cross-Clip Tracking Module** The proposed cross-clip tracking module iteratively stacks the axial-trajectory attention and Temporal-ASPP to refine all the clip object queries $\{\mathbf{C}_k\}_{k=1}^{K}$ of a video, obtaining a temporally consistent prediction at the video-level.

**Video-Level (Offline) Inference** With the proposed within-clip and cross-clip tracking modules, built on top of any clip-level video segmenter, we can now inference the whole video in an offline fashion by exploiting all the refined clip object queries. We first obtain the video features by concatenating all clip features produced by the pixel decoder (totally $K$ clips). The predicted video-level tubes are then generated by multiplying all the clip object queries with the video features (similar to image mask transformers (Wang et al., 2021a; Yu et al., 2022a)). To obtain the predicted classes for the video-level tubes, we exploit another 1D convolution layer (*i.e.*, the "Temporal 1D Conv" in the top-right of Fig. 5) to generate the temporally weighted class predictions, motivated by the fact that the object queries on the trajectory path should have the same class prediction.

Table 1: **Video Panoptic Segmentation (VPS) results.** We reproduce baseline Video-kMaX (column **RP**) by taking non-overlapping clips as input and replacing their hierarchical matching scheme with simple Hungarian Matching on object queries. We then compare our Axial-VS with other state-of-the-art works. Reported results of Axial-VS are averaged over 3 runs. **WC**: Our Within-Clip tracking module. **CC**: Our Cross-Clip tracking module.

a) **[VPS] Effects of proposed modules on VIPSeg val set**

| method | backbone | RP | WC | CC | VPQ | VPQ$^{Th}$ | VPQ$^{St}$ |
|---|---|---|---|---|---|---|---|
| Video-kMaX | ResNet50 | - | - | - | 38.2 | - | - |
| Video-kMaX | ResNet50 | ✓ | - | - | 42.7 | 42.5 | 42.9 |
| Axial-VS | ResNet50 | ✓ | ✓ | - | 46.1 | 45.6 | 46.6 |
| Axial-VS | ResNet50 | ✓ | ✓ | ✓ | 46.7 | 46.7 | 46.6 |
| Video-kMaX | ConvNeXt-L | - | - | - | 51.9 | - | - |
| Video-kMaX | ConvNeXt-L | ✓ | - | - | 52.7 | 54.1 | 51.3 |
| Axial-VS | ConvNeXt-L | ✓ | ✓ | - | 56.2 | 58.4 | 54.0 |
| Axial-VS | ConvNeXt-L | ✓ | ✓ | ✓ | 57.1 | 59.3 | 54.8 |
| Axial-VS | ConvNeXtV2-L | ✓ | ✓ | - | 57.7 | 58.3 | 57.1 |
| Axial-VS | ConvNeXtV2-L | ✓ | ✓ | ✓ | 58.0 | 58.8 | 57.2 |

b) **[VPS] Comparisons with others on VIPSeg val set**

| method | backbone | VPQ | VPQ$^{Th}$ | VPQ$^{St}$ |
|---|---|---|---|---|
| *online/near-online methods* | | | | |
| TarVIS (Athar et al., 2023) | ResNet50 | 33.5 | 39.2 | 28.5 |
| DVIS (Zhang et al., 2023) | ResNet50 | 39.2 | 39.3 | 39.0 |
| TarVIS (Athar et al., 2023) | Swin-L | 48.0 | 58.2 | 39.0 |
| DVIS (Zhang et al., 2023) | Swin-L | 54.7 | 54.8 | 54.6 |
| Axial-VS w/ Video-kMaX | ResNet50 | 46.1 | 45.6 | 46.6 |
| Axial-VS w/ Video-kMaX | ConvNeXt-L | 56.2 | 58.4 | 54.0 |
| Axial-VS w/ Video-kMaX | ConvNeXtV2-L | 57.7 | 58.3 | 57.1 |
| *offline methods* | | | | |
| DVIS (Zhang et al., 2023) | ResNet50 | 43.2 | 43.6 | 42.8 |
| DVIS (Zhang et al., 2023) | Swin-L | 57.6 | 59.9 | 55.5 |
| Axial-VS w/ Video-kMaX | ResNet50 | 46.7 | 46.7 | 46.6 |
| Axial-VS w/ Video-kMaX | ConvNeXt-L | 57.1 | 59.3 | 54.8 |
| Axial-VS w/ Video-kMaX | ConvNeXtV2-L | 58.0 | 58.8 | 57.2 |

## 4 Experimental Results

We evaluate Axial-VS based on two different clip-level segmenters on four widely used video segmentation benchmarks to show its generalizability. Specifically, for video panoptic segmentation (VPS), we build Axial-VS based on Video-kMaX (Shin et al., 2024) and report performance on VIPSeg (Miao et al., 2022). We also build Axial-VS on top of Tube-Link (Li et al., 2023b) for video instance segmentation (VIS) and report the performance on Youtube-VIS 2021 (Yang et al., 2021a), 2022 (Yang et al., 2022), and OVIS (Qi et al., 2022). Since Tube-Link is built on top of Mask2Former (Cheng et al., 2022) and thus already contains six layers of Multi-Scale Deformable Attention (MSDeformAttn), we simplify our within-clip tracking module by directly inserting axial-trajectory attention after each original MSDeformAttn. We follow the original setting of Video-kMaX and Tube-Link to use the same training losses. Note that when training the cross-clip tracking module, both the clip-level segmenter and the within-clip tracking module are frozen due to memory constraint. We utilize Video Panoptic Quality (VPQ), as defined in VPSNet (Kim et al., 2020), and Average Precision (AP), as defined in MaskTrack R-CNN (Yang et al., 2019), for evaluating the models on VPS and VIS, respectively. We provide more implementation details in the appendix.

### 4.1 Improvements over Baselines

We first provide a systematic study to validate the effectiveness of the proposed modules.

**Video Panoptic Segmentation (VPS)** Tab. 1a summarizes the improvements over the baseline Video-kMaX (Shin et al., 2024) on the VIPSeg dataset. For a fair comparison, we first reproduce Video-kMaX in our PyTorch framework (which was originally implemented in TensorFlow (Weber et al., 2021a)). Our re-implementation yields significantly improved VPQ results compared to the original model, with a 4.5% and 0.8% VPQ improvement using ResNet50 and ConvNeXt-L, respectively, establishing a solid baseline. As shown in the table, using the proposed within-clip tracking module improves over the reproduced solid baseline by 3.4% and 3.5% VPQ with ResNet50 and ConvNeXt-L, respectively. Employing the proposed cross-clip tracking module further improves the performance by additional 0.6% and 0.9% VPQ with ResNet50 and ConvNeXt-L, respectively. Finally, using the modern ConvNeXtV2-L brings another 1.5% and 0.9% improvements, when compared to the ConvNeXt-L counterparts.

**Video Instance Segmentation (VIS)** Tab. 2 summarizes the improvements over the baseline Tube-Link (Li et al., 2023b) on the Youtube-VIS-21, -22, and OVIS datasets. Similarly, for a fair comparison, we first reproduce the Tube-Link results, using their official code-base. Our reproduction yields similar performances to the original model, except OVIS, where we observe a gap of 4.1% AP for ResNet50. On Youtube-VIS-21 (Tab. 2a), the proposed within-clip tracking module improves the reproduced baselines by 0.6% and 0.6% for ResNet50 and Swin-L, respectively. Using our cross-clip tracking module additionally

Table 2: **Video Instance Segmentation (VIS) results.** We reproduce baseline Tube-Link (column **RP**) with their official code-base. We then build on top of it with our Within-Clip tracking module (**WC**) and Cross-Clip tracking module (**CC**). For Youtube-VIS-22, we mainly report $AP^{long}$ (for long videos) and see appendix for $AP^{short}$ (for short videos) and $AP^{all}$ (average of them). Reported results are averaged over 3 runs. §: Our best attempt to reproduce Tube-Link's performances (25.4%), lower than the results (29.5%) reported in the paper. Their provided checkpoints also yield lower results (26.7%). N/A: Not available from their code-base, but we have attempted to reproduce.

a) **[VIS] Youtube-VIS-21 val set**

| method | backbone | RP | WC | CC | AP |
|---|---|---|---|---|---|
| Tube-Link | ResNet50 | - | - | - | 47.9 |
| Tube-Link | ResNet50 | ✓ | - | - | 47.8 |
| Axial-VS | ResNet50 | ✓ | ✓ | - | 48.4 |
| Axial-VS | ResNet50 | ✓ | ✓ | ✓ | 48.5 |
| Tube-Link | Swin-L | - | - | - | 58.4 |
| Tube-Link | Swin-L | ✓ | - | - | 58.2 |
| Axial-VS | Swin-L | ✓ | ✓ | - | 58.8 |
| Axial-VS | Swin-L | ✓ | ✓ | ✓ | 59.1 |

b) **[VIS] Youtube-VIS-22 val set**

| method | backbone | RP | WC | CC | $AP^{long}$ |
|---|---|---|---|---|---|
| Tube-Link | ResNet50 | - | - | - | 31.1 |
| Tube-Link | ResNet50 | ✓ | - | - | 32.1 |
| Axial-VS | ResNet50 | ✓ | ✓ | - | 36.5 |
| Axial-VS | ResNet50 | ✓ | ✓ | ✓ | 37.0 |
| Tube-Link | Swin-L | - | - | - | 34.2 |
| Tube-Link | Swin-L | ✓ | - | - | 34.2 |
| Axial-VS | Swin-L | ✓ | ✓ | - | 35.9 |
| Axial-VS | Swin-L | ✓ | ✓ | ✓ | 38.9 |

c) **[VIS] OVIS val set**

| method | backbone | RP | WC | CC | AP |
|---|---|---|---|---|---|
| Tube-Link | ResNet50 | - | - | - | 29.5 |
| Tube-Link | ResNet50 | ✓ | - | - | 25.4§ |
| Axial-VS | ResNet50 | ✓ | ✓ | - | 27.6 |
| Axial-VS | ResNet50 | ✓ | ✓ | ✓ | 28.3 |
| Tube-Link | Swin-L | - | - | - | N/A |
| Tube-Link | Swin-L | ✓ | - | - | 33.3 |
| Axial-VS | Swin-L | ✓ | ✓ | - | 39.1 |
| Axial-VS | Swin-L | ✓ | ✓ | ✓ | 39.8 |

improves the performance by 0.1% and 0.3% for ResNet50 and Swin-L, respectively. On Youtube-VIS-22 (Tab. 2b), our proposed modules bring more significant improvements, showing our method's ability to handle the challenging long videos in the dataset. Specifically, using our within-clip tracking module shows 4.4% and 1.7% $AP^{long}$ for ResNet50 and Swin-L, respectively. Our cross-clip tracking module further improves the performances by 0.5% and 3.0% $AP^{long}$ for ResNet50 and Swin-L, respectively. On OVIS (Tab. 2c), even though we did not successfully reproduce Tube-Link (using their provided config files), we still observe a significant improvement brought by the proposed modules. Particularly, our within-clip tracking modules improves the baselines by 2.2% and 5.8% AP for ResNet50 and Swin-L, respectively. Another improvements of 0.7% and 0.7% AP for ResNet50 and Swin-L can be attained with the proposed cross-clip tracking module. To summarize, our proposed modules bring more remarkable improvements for long and challenging datasets.

## 4.2 Comparisons with Other Methods

After analyzing the improvements brought by the proposed modules, we now move on to compare our Axial-VS with other state-of-the-art methods.

**Video Panoptic Segmentation (VPS)** As shown in Tab. 1b, in the online/near-online setting, when using ResNet50, our Axial-VS significantly outperforms TarVIS (Athar et al., 2023) (which co-trains and exploits multiple video segmentation datasets) by a large margin of 12.6% VPQ. Axial-VS also outperforms the recent ICCV 2023 work DVIS (Zhang et al., 2023) by a healthy margin of 6.9% VPQ. When using the stronger backbones, Axial-VS with ConvNeXt-L still outperforms TarVIS and DVIS with Swin-L by 8.2% and 1.5% VPQ, respectively. The performance is further improved by using the modern ConvNeXtV2-L backbone, attaining 57.7% VPQ. In the offline setting, Axial-VS with ResNet50 outperforms DVIS by 3.5% VPQ, while Axial-VS with ConvNeXt-L performs comparably to DVIS with Swin-L. Finally, when using the modern ConvNeXtV2-L, Axial-VS achieves 58.0% VPQ, setting a new state-of-the-art.

**Video Instance Segmentation (VIS)** Tab. 3 compares Axial-VS with other state-of-the-art methods for VIS. On Youtube-VIS-21 (Tab. 3a), Axial-VS exhibits a slight performance advantage over TarVIS (Athar et al., 2023) and DVIS (Zhang et al., 2023) with an improvement of 0.1% and 1.1% AP, respectively. On Youtube-VIS-22 (Tab. 3b), Axial-VS outperforms DVIS in both online/near-online and offline settings by 5.3% and 1.1% $AP^{long}$, respectively.

## 4.3 Ablation Studies

We conduct ablation studies on VIPSeg, leveraging ResNet50 due to its scene diversity and long-length videos. Here, we present ablations on attention operations and cross-clip tracking design, as well as hyper-parameters such as the number of layers in the within-clip tracking and cross-clip tracking modules, clip length and sampling range. We further provide GFlops comparisons, more visualizations and failure cases in the appendix.

Table 3: **Video Instance Segmentation (VIS) results.** We compare our Axial-VS with other state-of-the-art works on Youtube-VIS-21 and Youtube-VIS-22 val set. Reported results of Axial-VS are averaged over 3 runs. *: All results are reproduced by us using their official checkpoints.

a) **[VIS] Youtube-VIS-21 val set**

| method | backbone | AP |
|---|---|---|
| *online/near-online methods* | | |
| TarVIS (Athar et al., 2023) | ResNet50 | 48.3 |
| Axial-VS | ResNet50 | 48.4 |
| *offline methods* | | |
| VITA (Heo et al., 2022) | ResNet50 | 45.7 |
| DVIS (Zhang et al., 2023) | ResNet50 | 47.4 |
| Axial-VS | ResNet50 | 48.5 |

b) **[VIS] Youtube-VIS-22 val set**

| method | backbone | AP$^{\text{long}}$ |
|---|---|---|
| *online/near-online methods* | | |
| DVIS (Zhang et al., 2023)* | ResNet50 | 31.2 |
| Axial-VS | ResNet50 | 36.5 |
| *offline methods* | | |
| VITA (Heo et al., 2022)* | ResNet50 | 31.9 |
| DVIS (Zhang et al., 2023)* | ResNet50 | 35.9 |
| Axial-VS | ResNet50 | 37.0 |

Table 4: **Ablations on attention operations in the within-clip tracking module and cross-clip tracking design.** For the within-clip tracking module, we compare Joint Space-Time Attention (Vaswani et al., 2017), Divided Space-Time Attention (Bertasius et al., 2021), Multi-Scale Deformable Attention (Zhu et al., 2020) (MSDeformAttn), Axial-Trajectory Attention, and TarVIS Temporal Neck (Athar et al., 2023) (*i.e.*, MSDeformAttn + Window Space-Time Attention). Visualizations are provided in Fig. 7 to illustrate the differences between the compared attentions. For the cross-clip tracking module, we compare VITA (Heo et al., 2022) and the proposed cross-clip tracking module. Reported results are averaged over 3 runs. −: Not using any operations. Our final setting is marked in grey.

a) **[Ablations] Attention in the within-clip tracking**

| attention operations | VPQ |
|---|---|
| - | 42.7 |
| Joint Space-Time Attn (Vaswani et al., 2017) | 43.2 |
| Divided Space-Time Attn (Bertasius et al., 2021) | 43.6 |
| MSDeformAttn (Zhu et al., 2020) | 44.5 |
| Axial-Trajectory Attn | 44.7 |
| MSDeformAttn + Window Space-Time Attn (Athar et al., 2023) | 44.9 |
| MSDeformAttn + Axial-Trajectory Attn | 46.1 |

b) **[Ablations] Cross-clip tracking design**

| cross-clip tracking design | video query | encoder | decoder | VPQ |
|---|---|---|---|---|
| - | - | - | - | 46.1 |
| VITA (Heo et al., 2022) | ✓ | ✓ | ✓ | 46.3 |
| cross-clip tracking module | ✗ | ✓ | ✗ | 46.7 |

**Attention Operations in Within-Clip Tracking Module**   In Tab. 4a, we ablate the attention operations used in the within-clip tracking module. To begin with, we utilize joint space-time attention (Vaswani et al., 2017), achieving a performance of 43.2% VPQ, which is a 0.5% improvement over the baseline. Subsequently, we apply divided space-time attention (Bertasius et al., 2021) (*i.e.*, decomposing space-time attention into space- and time-axes), resulting in a performance of 43.6% VPQ. This represents a further improvement of 0.4% VPQ over joint space-time attention, potentially due to its larger learning capacity, incorporating distinct learning parameters for temporal and spatial attention. Afterwards, we employ either only Multi-Scale Deformable Attention (Zhu et al., 2020) (MSDeformAttn) for spatial attention or only the proposed Axial-Trajectory Attention (AxialTrjAttn), sequentially along $H$- and $W$-axes for temporal attention, obtaining the performance of 44.5% and 44.7% VPQ, respectively. Replacing the attention operations with TarVIS Temporal Neck (Athar et al., 2023) (*i.e.*, MSDeformAttn + Window Space-Time Attention) increases the performance to 44.9% VPQ. Finally, if we change the attention scheme to the proposed MSDeformAttn + AxialTrjAttn, it brings another performance gain of 1.2% over TarVIS's design, achieving 46.1% VPQ. To better understand the distinctions among the four space-time self-attention schemes introduced above, we illustrate them in Fig. 7. On the temporal side, "Joint Space-Time Attention" simply attends to all pixels, while both "Divided Space-Time Attention" and "Window Space-Time Attention" focus on a fixed region across time. In contrast, our proposed *axial-trajectory attention* effectively tracks the object across time, capturing more accurate information and yielding more temporally consistent features.

**Tracking Design in Cross-Clip Tracking Module**   In Tab. 4b, we ablate the cross-clip tracking design in the cross-clip tracking module. We experiment with the design of VITA (Heo et al., 2022) to learn an additional set of video object queries by introducing a decoder for decoding information from encoded clip queries, yielding 46.3% VPQ, with a slight gain of 0.2% over the baseline. Replacing the VITA design with the proposed simple encoder-only and video-query-free design leads to a better performance of 46.7% VPQ.

**Within-Clip Tracking Module**   In Tab. 5, we ablate the design choices of the proposed within-clip tracking module. To begin with, we employ one MSDeformAttn and one TrjAttn (Trajectory Attention) with $N_w = 2$, obtaining the performance of 45.3% VPQ (+2.6% over the baseline). Replacing the TrjAttn

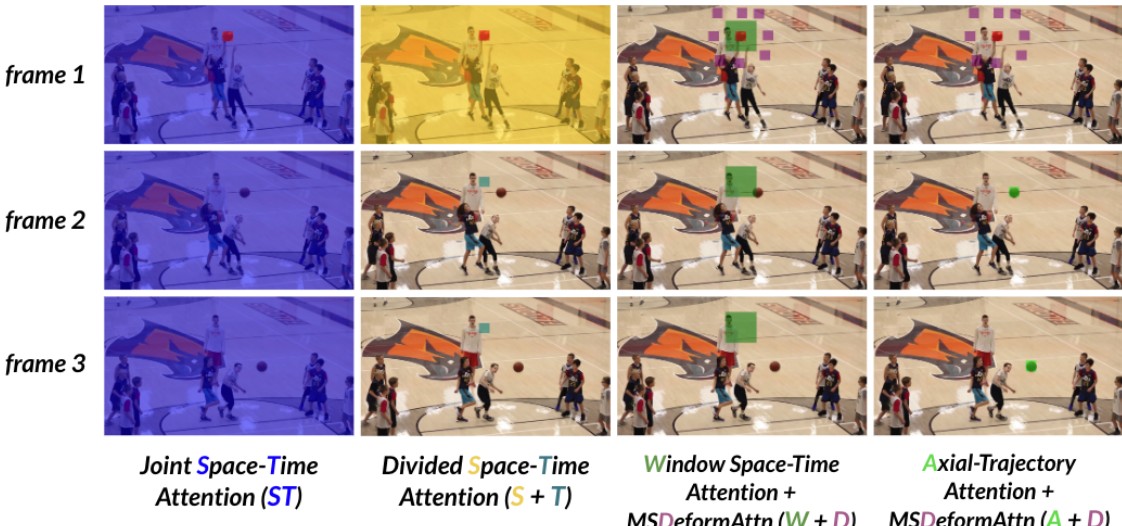

Figure 7: **Illustration of the four space-time self-attention schemes studied in this work.** For clarity, we represent the reference point at frame 1 in red and display its space-time attended pixels under each scheme in non-red colors. Pixels without color are not involved in the self-attention computation of the reference point. Different colors within a scheme represent attentions applied along distinct dimensions. Note that the visualizations of Multi-Scale Deformable Attention (MSDeformAttn) and Axial-Trajectory Attention are simplified for improving visual clarity.

Table 5: **Ablation on within-clip tracking module.** We vary the number of Multi-Scale Deformable Attention (#MSDeformAttn), number of Trajectory Attention (#TrjAttn), or number of Axial-Trajectory Attention (#AxialTrjAttn). $N_w$ denotes the number of blocks (*i.e.*, repetitions). Numbers are averaged over 3 runs. −: Not using any operations. N/A: Not avaliable due to insufficient GPU memory capacity. The final setting is marked in grey.

| #MSDeformAttn | #TrjAttn | #AxialTrjAttn | $N_w$ | VPQ |
|:---:|:---:|:---:|:---:|:---|
| - | - | - | - | 42.7 |
| 1 | 1 | - | 2 | 45.3 |
| 1 | 2 | - | 2 | N/A |
| 1 | - | 1 | 2 | 45.4 |
| 1 | - | 2 | 1 | 44.7 |
| 1 | - | 2 | 2 | 46.1 |
| 1 | - | 2 | 3 | 45.2 |
| 1 | - | 3 | 2 | 45.7 |
| 1 | - | 4 | 2 | 45.5 |
| 2 | - | 4 | 1 | 45.8 |

with AxialTrjAttn yields a comparable performance of 45.4%. We note that it will be Out-Of-Memory, if we stack two TrjAttn layers in a V100 GPU. Stacking two AxialTrjAttn layers in each block leads to our final setting with 46.1%. Increasing or decreasing the number of blocks $N_w$ degrades the performance slightly. If we employ one more AxialTrjAttn layers per block, the performance drops by 0.4%. Finally, if we change the iterative stacking scheme to a sequential manner (*i.e.*, stacking two MSDeformAttn, followed by four AxialTrjAttn), the performance also decreases slightly by 0.3%.

**Cross-Clip Tracking Module** Tab. 6 summarizes our ablation studies on the design choices of the proposed cross-clip tracking module. Particularly, in Tab. 6a, we adopt different operations in the module. Using self-attention (SelfAttn), instead of axial-trajectory attention (AxialTrjAttn) degrades the performance by 0.3% VPQ. Removing the Temporal-ASPP operation also decreases the performance by 0.2%. In Tab. 6b, we ablate the atrous rates used in the three parallel temporal convolutions of the proposed Temporal-ASPP. Using atrous rates $(1, 2, 3)$ (*i.e.*, rates set to 1, 2, and 3 for those three convolutions, respectively) leads to the best performance. In Tab. 6c, we find that using $N_c = 4$ blocks in the cross-clip tracking module yields the best result.

Table 6: **Ablation on cross-clip tracking module.** We vary operations in the block, Temporal-ASPP (atrous rates), and number of blocks $N_c$. Numbers are averaged over 3 runs. The final setting is marked in grey.

a) **Operations**

| SelfAttn | AxialTrjAttn | Temporal-ASPP | VPQ |
|:---:|:---:|:---:|:---|
| ✓ | | ✓ | 46.4 |
| | ✓ | ✓ | 46.7 |
| | ✓ | | 46.5 |

b) **Temporal-ASPP**

| atrous rates | VPQ |
|:---:|:---|
| (1, 2, 3) | 46.7 |
| (1, 2, 5) | 46.5 |
| (1, 3, 5) | 46.4 |

c) **Number of blocks** $N_c$

| $N_c$ | VPQ |
|:---:|:---|
| 4 | 46.7 |
| 6 | 46.7 |
| 8 | 46.2 |

Table 7: **Ablation on clip length and clip sampling range.** We vary the clip length $T$ and sampling range (*i.e.*, frame index interval) of a clip. Numbers are averaged over 3 runs. The final setting is marked in grey.

a) **Clip length**

| clip length | Video-kMaX | Axial-VS (near-online) | Axial-VS (offline) |
|:---:|:---:|:---:|:---:|
| 2 | 42.7 | 46.1 | 46.7 |
| 3 | 42.1 | 45.1 | 45.5 |
| 4 | 41.4 | 44.2 | 44.7 |

b) **Clip sampling range**

| range | Axial-VS (near-online) |
|:---:|:---:|
| ±1 | 46.1 |
| ±3 | 45.8 |
| ±10 | 43.9 |

**Clip Length and Clip Sampling Range** Tab. 7 summarizes our ablation studies on the clip length $T$ (*i.e.*, number of frames in a clip) and clip sampling range (*i.e.*, frame index interval). Concretely, in Tab. 7a, we adopt different clip sizes for training the segmenters. As observed, the performance of Video-kMaX (Shin et al., 2024) gradually decreases with the increase of clip size ($42.7\% \rightarrow 42.1\% \rightarrow 41.4\%$). However, both our Axial-VS near-online (with within-clip tracking module) and Axial-VS offline (with within-clip + cross-clip tracking module) models consistently bring steady improvements. Specifically, for a clip size of 2, the within-clip tracking module enhances Video-kMaX performance by 3.4%, achieving 46.1% VPQ. Subsequently, our cross-clip tracking module further elevates the performance by 0.6% to 46.7% VPQ. These improvements are also notable for other clip sizes. In conclusion, the observed performance drops are primarily influenced by the performance variance of the deployed clip-level segmenters (i.e., Video-kMaX). We propose two main hypotheses: Firstly, in existing video panoptic segmentation datasets such as VIPSeg, objects typically exhibit slow movement. Therefore, neighboring frames contain the most informative data, with minimal additional information gained from including more frames. Secondly, the transformer decoders employed in Video-kMaX may encounter challenges when processing larger feature maps associated with longer clip lengths. As indicated in their original paper, Video-kMaX also adopts a clip length of 2 in their final settings when training on VIPSeg. In Tab. 7b, we ablate the sampling range used when constructing a training clip. Using continuous frames (*i.e.*, ±1) leads to the best performance. While slightly increasing the sampling range to ±3 degrades the performance slightly by 0.3% to 45.8% VPQ, increasing it to ±10 greatly hampers the learning of the within-clip tracking module, yielding only 43.9% VPQ.

## 5 Conclusion

In conclusion, our contribution, Axial-VS, represents a meta-architecture that elevates the capabilities of a standard clip-level segmenter through the incorporation of within-clip and cross-clip tracking modules. These modules, empowered by axial-trajectory attention, strategically enhance short-term and long-term temporal consistency. The exemplary performance of Axial-VS on video segmentation benchmarks underscores its efficacy in mitigating the limitations observed in contemporary clip-level video segmenters.

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

## Appendix

In the appendix, we provide additional information as listed below:

- Sec. A provides the implementation details.

- Sec. B provides additional experimental results, including computational cost (GFLOPs), running time (FPS) and memory consumption (VRAM) comparison and more comparison with other methods for video panoptic segmentation (VPS) and video instance segmentation (VIS).

- Sec. C provides prediction visualizations and additional axial-trajectory attention visualization results.

- Sec. D discusses our method's limitations.

- Sec. E provides the dataset information.

- Sec. F discusses the broader impact of Axial-VS.

## A    Implementation Details

**Implementation Details**   The proposed Axial-VS is a unified approach for both near-online and offline video segmentation (*i.e.*, the cross-clip tracking module is only used for the offline setting). For the near-online setting (*i.e.*, employing the within-clip tracking module), we use a clip size of two and four for VPS and VIS, respectively. For the offline setting (*i.e.*, employing the cross-clip tracking module), we adopt a video length of 24 (*i.e.*, 12 clips) for VPS and 20 (*i.e.*, 5 clips) for VIS. At this stage, we only train the cross-clip tracking module, while both the clip-level segmenter and the within-clip tracking module are frozen due to memory constraint. During testing, we directly inference with the whole video with our full model.

We experiment with four backbones for Axial-VS: ResNet50 (He et al., 2016), Swin-L (Liu et al., 2021), ConvNeXt-L (Liu et al., 2022b) and ConvNeXt V2-L (Woo et al., 2023). For VPS experiments, we first reproduce Video-kMaX (Shin et al., 2024) based on the official PyTorch re-implementation of kMaX-DeepLab (Yu et al., 2022b). We employ a specific pre-training protocol for VIPSeg, closely following the prior works (Weber et al., 2021b; Kim et al., 2022; Shin et al., 2024). Concretely, starting with an ImageNet (Russakovsky et al., 2015) pre-trained backbone, we pre-train the kMaX-DeepLab and Multi-Scale Deformable Attention (MSDeformAttn) in our within-clip tracking module on COCO (Lin et al., 2014). The within-clip and cross-clip tracking modules deploy $N_w = 2$ and $N_c = 4$ blocks, respectively, for VPS. On the other hand, for VIS experiments, we use the official code-base of Tube-Link (Li et al., 2023b). Since Tube-Link is built on top of Mask2Former (Cheng et al., 2022) and thus already contains six layers of MSDeformAttn, we simplify our within-clip tracking module by directly inserting axial-trajectory attention after each original MSDeformAttn. As a result, the within-clip and cross-clip tracking modules use $N_w = 6$ and $N_c = 4$ blocks, respectively, for VIS. We note that we do not use any other video datasets (*e.g.*, pseudo COCO videos) for pre-training axial-trajectory attention.

We closely adhere to the training protocols established by the baseline clip-level segmenters. Specifically, for the VPS task with ResNet50 as the backbone, we adopt the training methodology of Video-kMaX. Our near-online Axial-VS is trained on the VIPSeg dataset with a clip size of $2 \times 769 \times 1345$ and a batch size of 32, utilizing 16 V100 32G GPUs for 40k iterations. This training regimen spans approximately 13 hours. Additionally, our offline Axial-VS is trained on VIPSeg with a video size of $24 \times 769 \times 1345$ (12 clips, each comprising 2 frames) and a batch size of 16, employing 8 A100 80G GPUs for 15k iterations. This training process requires approximately 10 hours. For the VIS task with ResNet50 as the backbone, we adopt the Tube-Link training protocol. Our near-online Axial-VS is trained on Youtube-VIS with a batch size of 8 clips (each containing 4 frames) using 8 V100 32G GPUs for 15k iterations. We adhere to the literature by randomly resizing the shortest edge of each clip to a predetermined size within the range [288, 320, 352, 384, 416, 448, 480, 512]. This training process takes approximately 7 hours. Additionally, our offline Axial-VS is trained on Youtube-VIS with a batch size of 8 videos (each comprising 20 frames, equivalent to 5 clips) using 8 V100 32G GPUs for 10k iterations. This training process requires approximately 4 hours.

Table 8: **GFlops, FPS and VRAM comparisons on attention operations in the within-clip tracking module.** We compare Joint Space-Time Attention (Vaswani et al., 2017), Divided Space-Time Attention (Bertasius et al., 2021), Multi-Scale Deformable Attention (Zhu et al., 2020) (MSDeformAttn), the proposed Axial-Trajectory Attention, and TarVIS Temporal Neck (Athar et al., 2023) (*i.e.*, MSDeformAttn + Window Space-Time Attention). The GFLOPs and FPS are obtained by measuring on models with ResNet50 as the backbone using an A100 GPU. We report the VRAM on two different input resolutions: $2 \times 513 \times 897$ and $2 \times 769 \times 1345$, respectively. Reported results are averaged over 3 runs. $-$: Not using any operations. Our final setting is marked in grey.

| | input clip resolution | | | | |
| | $2 \times 513 \times 897$ | $2 \times 769 \times 1345$ | | | |
| attention operations | VRAM | GFLOPs | FPS | VRAM | VPQ |
|---|---|---|---|---|---|
| - | 6.74G | 354 | 14.3 | 11.90G | 42.7 |
| Joint Space-Time Attn (Vaswani et al., 2017) | 9.87G | 493 | 10.3 | 25.97G | 43.2 |
| Divided Space-Time Attn (Bertasius et al., 2021) | 8.58G | 430 | 12.6 | 19.58G | 43.6 |
| MSDeformAttn (Zhu et al., 2020) | 7.75G | 432 | 12.5 | 14.15G | 44.5 |
| Axial-Trajectory Attn | 7.57G | 443 | 11.7 | 13.81G | 44.7 |
| MSDeformAttn + Window Space-Time Attn (Athar et al., 2023) | 8.21G | 476 | 10.5 | 15.15G | 44.9 |
| MSDeformAttn + Axial-Trajectory Attn | 8.38G | 481 | 10.5 | 15.59G | 46.1 |

Table 9: **GFLOPs comparisons on cross-clip tracking design.** For the cross-clip tracking module, we compare VITA (Heo et al., 2022) and the proposed cross-clip tracking module and report GFLOPs of them only. Reported results are averaged over 3 runs. $-$: Not using any operations. Our final setting is marked in grey.

| cross-clip tracking design | video query | encoder | decoder | GFLOPs | VPQ |
|---|---|---|---|---|---|
| - | - | - | - | - | 46.1 |
| VITA (Heo et al., 2022) | ✓ | ✓ | ✓ | 47 | 46.3 |
| cross-clip tracking module | ✗ | ✓ | ✗ | 32 | 46.7 |

# B   Additional Experimental Results

In this section, we provide more experimental results, including the computational cost (GFLOPs), running time (FPS) and memory consumption (VRAM) comparisons on the proposed within-clip tracking module along with GFlops comparisons on the cross-clip tracking modules (Sec. B.1), as well as more detailed comparisons with other state-of-the-art methods (Sec. B.2).

## B.1   GFLOPs, FPS and VRAM Comparisons

We conduct the GFLOPs, FPS and VRAM comparisons on the VIPSeg dataset, using ResNet50.

**GFLOPs, FPS and VRAM Comparisons on Attention Operations in Within-Clip Tracking Module**   In Tab. 8, we present a comparison of the GFLOPs, FPS and VRAM for the attention operations used in the within-clip tracking module. The GFlops and FPS are evaluated using a short clip of size $2 \times 769 \times 1345$ on an A100 GPU. Additionally, VRAM is reported for two different input clip resolutions: $2 \times 513 \times 897$ and $2 \times 769 \times 1345$, respectively. The table highlights that the proposed "Axial-Trajectory Attn" introduces a moderate increase in GFlops and VRAM, along with a modest decrease in FPS, while significantly enhancing performance in VPQ. Notably, "Divided Space-Time Attn", "MSDeformAttn", and "Axial-Trajectory Attn" exhibit comparable computational costs and GPU memory usage. Conversely, "Joint Space-Time Attn" imposes the highest computational load and GPU memory consumption due to its compute-intensive attention operations on high-resolution dense pixel feature maps.

**GFLOPs Comparisons on Tracking Design in Cross-Clip Tracking Module**   In Tab. 9, we present a comparison of the GFLOPs for the cross-clip tracking design. The numbers are computed using a video of size $24 \times 769 \times 1345$, specifically measuring the computational costs of the cross-clip tracking module. The table shows that our cross-clip tracking module is more lightweight compared to VITA, yet achieves superior performance, which can be attributed to the simple and effective design of our cross-clip tracking module.

Table 10: **VIPSeg *val* set results.** We provide more complete comparisons with other state-of-the-art methods. Numbers of Axial-VS are averaged over 3 runs. ‡: Evaluated using their open-source checkpoint.

| method | backbone | VPQ | VPQ$^{Th}$ | VPQ$^{St}$ |
|---|---|---|---|---|
| *online/near-online methods* | | | | |
| ViP-DeepLab (Qiao et al., 2021) | ResNet50 | 16.0 | - | - |
| VPSNet-FuseTrack (Kim et al., 2020) | ResNet50 | 17.0 | - | - |
| VPSNet-SiamTrack (Woo et al., 2021) | ResNet50 | 17.2 | - | - |
| Clip-PanoFCN (Miao et al., 2022) | ResNet50 | 22.9 | - | - |
| Video K-Net (Li et al., 2022) | ResNet50 | 26.1 | - | - |
| TubeFormer (Kim et al., 2022) | Axial-ResNet50-B3 | 31.2 | - | - |
| TarVIS (Athar et al., 2023) | ResNet50 | 33.5 | 39.2 | 28.5 |
| Video-kMaX (Shin et al., 2024) | ResNet50 | 38.2 | - | - |
| Tube-Link (Li et al., 2023b) | ResNet50 | 39.2 | - | - |
| DVIS (Zhang et al., 2023)‡ | ResNet50 | 39.2 | 39.3 | 39.0 |
| TarVIS (Athar et al., 2023) | Swin-L | 48.0 | 58.2 | 39.0 |
| Video-kMaX (Shin et al., 2024) | ConvNeXt-L | 51.9 | - | - |
| DVIS (Zhang et al., 2023) | Swin-L | 54.7 | 54.8 | 54.6 |
| Axial-VS w/ Video-kMaX (ours) | ResNet50 | 46.1 | 45.6 | 46.6 |
| Axial-VS w/ Video-kMaX (ours) | ConvNeXt-L | 56.2 | 58.4 | 54.0 |
| Axial-VS w/ Video-kMaX (ours) | ConvNeXt V2-L | 57.7 | 58.3 | 57.1 |
| *offline methods* | | | | |
| DVIS (Zhang et al., 2023) | ResNet50 | 43.2 | 43.6 | 42.8 |
| DVIS (Zhang et al., 2023) | Swin-L | 57.6 | 59.9 | 55.5 |
| Axial-VS w/ Video-kMaX (ours) | ResNet50 | 46.7 | 46.7 | 46.6 |
| Axial-VS w/ Video-kMaX (ours) | ConvNeXt-L | 57.1 | 59.3 | 54.8 |
| Axial-VS w/ Video-kMaX (ours) | ConvNeXt V2-L | 58.0 | 58.8 | 57.2 |

## B.2 Comparisons with Other Methods

**Video Panoptic Segmentation (VPS)** In Tab. 10, we compare with more state-of-the-art methods on the VIPSeg dataset. We observe the similar trend as discussed in the main paper, and thus simply list all the other methods for a complete comparison.

**Video Instance Segmentation (VIS)** In Tab. 11, we report more state-of-the-art methods on the Youtube-VIS-21 dataset. As shown in the table, our Axial-VS with ResNet50 backbone demonstrates a better performance than the other methods as discussed in the main paper, while our Axial-VS with Swin-L performs slightly worse than TarVIS (Athar et al., 2023) in the online/near-online setting and than DVIS (Zhang et al., 2023) in the offline setting. We think the performance can be improved by exploiting more video segmentation datasets, as TarVIS did, or by improving the clip-level segmenter. Particularly, our baseline Tube-Link with Swin-L performs worse than the other state-of-the-art methods with Swin-L.

For the Youtube-VIS-22 results, we notice that the reported numbers in some recent papers are not comparable, since some papers report AP$^{long}$ (AP for long videos) while some papers use AP$^{all}$, which is the average of AP$^{long}$ and AP$^{short}$ (AP for short videos). To carefully and fairly compare between methods, we therefore reproduce all the state-of-the-art results by using their official open-source checkpoints, and clearly report their AP$^{all}$, AP$^{long}$, and AP$^{short}$ in Tab. 12. Similar to the discussion in the main paper, our Axial-VS with ResNet50 significantly improves over the baseline Tube-Link and performs better than other state-of-the-art methods, particularly in AP$^{long}$. However, our results with Swin-L lag behind other state-of-the-art methods with Swin-L, whose gap may be bridged by improving the baseline Tube-Link Swin-L.

In Tab. 13, we summarize more comparisons with other state-of-the-art methods on OVIS. As shown in the table, our method remarkably improves over the baseline, but performs worse than the state-of-the-art methods, partially because we fail to fully reproduce the baseline Tube-Link that our method heavily depends upon. Similar to our other VIS results, we think the improvement of clip-level segmenter will also lead to the improvement of Axial-VS.

Table 11: **Youtube-VIS-21 *val* set results.** We provide more complete comparisons with other state-of-the-art methods. Numbers of Axial-VS are averaged over 3 runs.

| method | backbone | AP | $AP_{50}$ | $AP_{75}$ | $AR_1$ | $AR_{10}$ |
|---|---|---|---|---|---|---|
| *online/near-online methods* | | | | | | |
| MinVIS (Huang et al., 2022) | ResNet50 | 44.2 | 66.0 | 48.1 | 39.2 | 51.7 |
| IDOL (Wu et al., 2022c) | ResNet50 | 43.9 | 68.0 | 49.6 | 38.0 | 50.9 |
| GenVIS$_{near-online}$ (Heo et al., 2023) | ResNet50 | 46.3 | 67.0 | 50.2 | 40.6 | 53.2 |
| DVIS (Zhang et al., 2023) | ResNet50 | 46.4 | 68.4 | 49.6 | 39.7 | 53.5 |
| GenVIS$_{online}$ (Heo et al., 2023) | ResNet50 | 47.1 | 67.5 | 51.5 | 41.6 | 54.7 |
| Tube-Link (Li et al., 2023b) | ResNet50 | 47.9 | 70.0 | 50.2 | 42.3 | 55.2 |
| TarVIS (Athar et al., 2023) | ResNet50 | 48.3 | 69.6 | 53.2 | 40.5 | 55.9 |
| MinVIS (Huang et al., 2022) | Swin-L | 55.3 | 76.6 | 62.0 | 45.9 | 60.8 |
| IDOL (Wu et al., 2022c) | Swin-L | 56.1 | 80.8 | 63.5 | 45.0 | 60.1 |
| Tube-Link (Li et al., 2023b) | Swin-L | 58.4 | 79.4 | 64.3 | 47.5 | 63.6 |
| DVIS (Zhang et al., 2023) | Swin-L | 58.7 | 80.4 | 66.6 | 47.5 | 64.6 |
| GenVIS$_{online}$ (Heo et al., 2023) | Swin-L | 59.6 | 80.9 | 65.8 | 48.7 | 65.0 |
| GenVIS$_{near-online}$ (Heo et al., 2023) | Swin-L | 60.1 | 80.9 | 66.5 | 49.1 | 64.7 |
| TarVIS (Athar et al., 2023) | Swin-L | 60.2 | 81.4 | 67.6 | 47.6 | 64.8 |
| Axial-VS w/ Tube-Link (ours) | ResNet50 | 48.4 | 71.1 | 51.8 | 42.0 | 57.4 |
| Axial-VS w/ Tube-Link (ours) | Swin-L | 58.8 | 81.3 | 65.0 | 46.7 | 62.7 |
| *offline methods* | | | | | | |
| VITA (Heo et al., 2022) | ResNet50 | 45.7 | 67.4 | 49.5 | 40.9 | 53.6 |
| DVIS (Zhang et al., 2023) | ResNet50 | 47.4 | 71.0 | 51.6 | 39.9 | 55.2 |
| VITA (Heo et al., 2022) | Swin-L | 57.5 | 80.6 | 61.0 | 47.7 | 62.6 |
| DVIS (Zhang et al., 2023) | Swin-L | 60.1 | 83.0 | 68.4 | 47.7 | 65.7 |
| Axial-VS w/ Tube-Link (ours) | ResNet50 | 48.5 | 70.9 | 52.4 | 42.3 | 57.9 |
| Axial-VS w/ Tube-Link (ours) | Swin-L | 59.1 | 81.9 | 64.9 | 46.9 | 63.8 |

Table 12: **Youtube-VIS-22 *val* set results.** We provide more complete comparisons with other state-of-the-art methods. Numbers of Axial-VS are averaged over 3 runs. *: All results are reproduced by us using their official checkpoints. We report $AP^{short}$ and $AP^{long}$ for short and long videos, respectively, and $AP^{all}$ by averaging them.

| method | backbone | $AP^{all}$ | $AP^{short}$ | $AP_{50}$ | $AP_{75}$ | $AR_1$ | $AR_{10}$ | $AP^{long}$ | $AP_{50}$ | $AP_{75}$ | $AR_1$ | $AR_{10}$ |
|---|---|---|---|---|---|---|---|---|---|---|---|---|
| *online/near-online methods* | | | | | | | | | | | | |
| MinVIS (Huang et al., 2022)* | ResNet50 | 32.8 | 43.9 | 66.9 | 47.5 | 38.8 | 51.9 | 21.6 | 42.9 | 18.1 | 18.8 | 25.6 |
| GenVIS$_{near-online}$ (Heo et al., 2023)* | ResNet50 | 38.1 | 45.9 | 66.3 | 50.2 | 40.8 | 53.7 | 30.3 | 50.9 | 32.7 | 25.5 | 36.2 |
| DVIS (Zhang et al., 2023)* | ResNet50 | 38.6 | 46.0 | 68.1 | 50.4 | 39.7 | 53.5 | 31.2 | 50.4 | 36.8 | 30.2 | 35.7 |
| Tube-Link (Li et al., 2023b)* | ResNet50 | 39.5 | 47.9 | 70.4 | 50.5 | 42.6 | 55.9 | 31.1 | 56.1 | 31.2 | 29.1 | 36.3 |
| MinVIS (Huang et al., 2022)* | Swin-L | 43.5 | 55.0 | 77.8 | 60.6 | 45.3 | 60.3 | 31.9 | 51.4 | 33.0 | 28.2 | 35.3 |
| Tube-Link (Li et al., 2023b)* | Swin-L | 46.0 | 57.8 | 78.7 | 63.4 | 47.0 | 62.7 | 34.2 | 53.2 | 37.9 | 31.5 | 38.9 |
| DVIS (Zhang et al., 2023)* | Swin-L | 48.9 | 58.8 | 80.6 | 65.9 | 47.5 | 63.9 | 39.0 | 56.0 | 43.0 | 33.0 | 43.5 |
| Axial-VS w/ Tube-Link (ours) | ResNet50 | 41.6 | 46.8 | 68.1 | 50.5 | 41.5 | 56.2 | 36.5 | 61.1 | 41.7 | 32.3 | 42.3 |
| Axial-VS w/ Tube-Link (ours) | Swin-L | 47.3 | 58.7 | 81.1 | 64.9 | 46.9 | 62.7 | 35.9 | 62.0 | 37.0 | 34.2 | 39.7 |
| *offline methods* | | | | | | | | | | | | |
| VITA (Heo et al., 2022)* | ResNet50 | 38.8 | 45.7 | 66.6 | 50.1 | 41.0 | 53.1 | 31.9 | 53.8 | 37.0 | 31.1 | 37.3 |
| DVIS (Zhang et al., 2023)* | ResNet50 | 41.6 | 47.2 | 70.8 | 51.0 | 40.0 | 54.9 | 35.9 | 58.4 | 39.9 | 32.2 | 41.9 |
| VITA (Heo et al., 2022)* | Swin-L | 49.3 | 57.6 | 80.4 | 62.5 | 47.7 | 62.3 | 41.0 | 62.1 | 43.9 | 39.4 | 43.5 |
| DVIS (Zhang et al., 2023)* | Swin-L | 52.4 | 59.9 | 82.7 | 68.3 | 47.8 | 65.2 | 44.9 | 66.3 | 48.9 | 37.1 | 53.2 |
| Axial-VS w/ Tube-Link (ours) | ResNet50 | 41.3 | 45.6 | 68.0 | 51.1 | 40.2 | 54.7 | 37.0 | 63.4 | 36.7 | 29.0 | 40.2 |
| Axial-VS w/ Tube-Link (ours) | Swin-L | 48.8 | 58.7 | 81.0 | 64.2 | 46.6 | 63.5 | 38.9 | 64.4 | 39.3 | 32.0 | 42.3 |

## C  Visualization Results

**Visualizations of Prediction**  We provide visualization results in Fig. 8, Fig. 9, Fig. 10, and Fig. 11 for different video sequences. We compare with DVIS (Zhang et al., 2023) and our re-implemented Video-kMaX (Shin et al., 2024) with ResNet50 as backbone and inference in an online/near-online fashion.

**Visualizations of Learned Axial-Trajectory Attention**  We provide more visualizations of the learned axial-trajectory attention maps in Fig. 12, Fig. 13 and 14. Concretely, in Fig. 12, we illustrate the feasibility

Table 13: **OVIS *val* set results.** We provide more complete comparisons with other state-of-the-art methods. Numbers of Axial-VS are averaged over 3 runs. [§]: Reproduced by us using their official code-base.

| method | backbone | AP | $AP_{50}$ | $AP_{75}$ | $AR_1$ | $AR_{10}$ |
|---|---|---|---|---|---|---|
| *online/near-online methods* | | | | | | |
| MinVIS (Huang et al., 2022) | ResNet50 | 25.0 | 45.5 | 24.0 | 13.9 | 29.7 |
| Tube-Link (Li et al., 2023b)[§] | ResNet50 | 25.4 | 44.9 | 26.5 | 14.1 | 30.1 |
| Tube-Link (Li et al., 2023b) | ResNet50 | 29.5 | 51.5 | 30.2 | 15.5 | 34.5 |
| IDOL (Wu et al., 2022c) | ResNet50 | 30.2 | 51.3 | 30.0 | 15.0 | 37.5 |
| DVIS (Zhang et al., 2023) | ResNet50 | 30.2 | 55.0 | 30.5 | 14.5 | 37.3 |
| TarVIS (Athar et al., 2023) | ResNet50 | 31.1 | 52.5 | 30.4 | 15.9 | 39.9 |
| GenVIS$_{near-online}$ (Heo et al., 2023) | ResNet50 | 34.5 | 59.4 | 35.0 | 16.6 | 38.3 |
| GenVIS$_{online}$ (Heo et al., 2023) | ResNet50 | 35.8 | 60.8 | 36.2 | 16.3 | 39.6 |
| Tube-Link (Li et al., 2023b)[§] | Swin-L | 33.3 | 54.6 | 32.8 | 16.8 | 37.7 |
| MinVIS (Huang et al., 2022) | Swin-L | 39.4 | 61.5 | 41.3 | 18.1 | 43.3 |
| IDOL (Wu et al., 2022c) | Swin-L | 42.6 | 65.7 | 45.2 | 17.9 | 49.6 |
| TarVIS (Athar et al., 2023) | Swin-L | 43.2 | 67.8 | 44.6 | 18.0 | 50.4 |
| GenVIS$_{online}$ (Heo et al., 2023) | Swin-L | 45.2 | 69.1 | 48.4 | 19.1 | 48.6 |
| GenVIS$_{near-online}$ (Heo et al., 2023) | Swin-L | 45.4 | 69.2 | 47.8 | 18.9 | 49.0 |
| DVIS (Zhang et al., 2023) | Swin-L | 47.1 | 71.9 | 49.2 | 19.4 | 52.5 |
| Axial-VS w/ Tube-Link | ResNet50 | 27.6 | 50.1 | 27.2 | 14.6 | 32.5 |
| Axial-VS w/ Tube-Link | Swin-L | 39.1 | 62.3 | 39.8 | 18.5 | 42.3 |
| *offline methods* | | | | | | |
| VITA (Heo et al., 2022) | ResNet50 | 19.6 | 41.2 | 17.4 | 11.7 | 26.0 |
| DVIS (Zhang et al., 2023) | ResNet50 | 33.8 | 60.4 | 33.5 | 15.3 | 39.5 |
| VITA (Heo et al., 2022) | Swin-L | 27.7 | 51.9 | 24.9 | 14.9 | 33.0 |
| DVIS (Zhang et al., 2023) | Swin-L | 48.6 | 74.7 | 50.5 | 18.8 | 53.8 |
| Axial-VS w/ Tube-Link | ResNet50 | 28.3 | 50.7 | 27.0 | 14.6 | 34.0 |
| Axial-VS w/ Tube-Link | Swin-L | 39.8 | 64.5 | 40.1 | 17.9 | 43.7 |

of decomposing object motion into height- and width-axis components. We select the football in the first frame as the *reference point* and show the height and width axial-trajectory attention separately. We then multiply the height and width axial-trajectory attentions to visualize the trajectory of the reference point over time. In Fig. 13, we select the basketball in the first frame as the *reference point* and show that our axial-trajectory attention accurately tracks it along the moving trajectory. In Fig. 14, we select the black table in the first frame as the *reference point*. We note that the camera motion is very small in this short clip and the table thus remains static. Our axial-trajectory attention still accurately keeps tracking at the same location as time goes by.

**Failure Cases for Prediction**   We provide visualizations of failure cases of Axial-VS in Fig. 15 and 16. In general, we observe three common patterns of errors: heavy occlusion, fast moving objects, and extreme illumination. Specifically, the first challenge is that when there are heavy occlusions caused by multiple close-by instances, Axial-VS suffers from ID switching, leading Axial-VS to assign inconsistent ID to the same instance. For example, in clip (a) of Fig. 15, the ID of the human in red dress changes between frame 2 and 3, while in clip (b) of Fig. 15 the two humans in the back are recognized as only one human until frame 3 due to the heavy occlusion. The second common error is that in videos containing fast motion, Axial-VS suffers from precisely predicting the boundary of the moving object. In clip (c) of Fig. 16, the human's legs are not segmented out in frame 1 and 3. The last common error is that in videos containing extreme or varying illumination, Axial-VS might fail to detect the objects thus fails to generate consistent segmentation. In clip (d) of Fig. 16, the objects under the extreme illumination can not be well segmented.

**Failure Cases for Learned Axial-Trajectory Attention**   In Fig. 17 and 18, we show two failure cases of axial-trajectory attention where the selected *reference point* is not discriminative enough, sometimes yielding inaccurate axial-trajectory. To be specific, in Fig. 17, we select the left light of the subway in the first frame as *reference point*. Though axial-trajectory attention precisely associates its position at the second frame, in the third frame the attention becomes sparse, mostly because that there are many similar 'light' objects in the

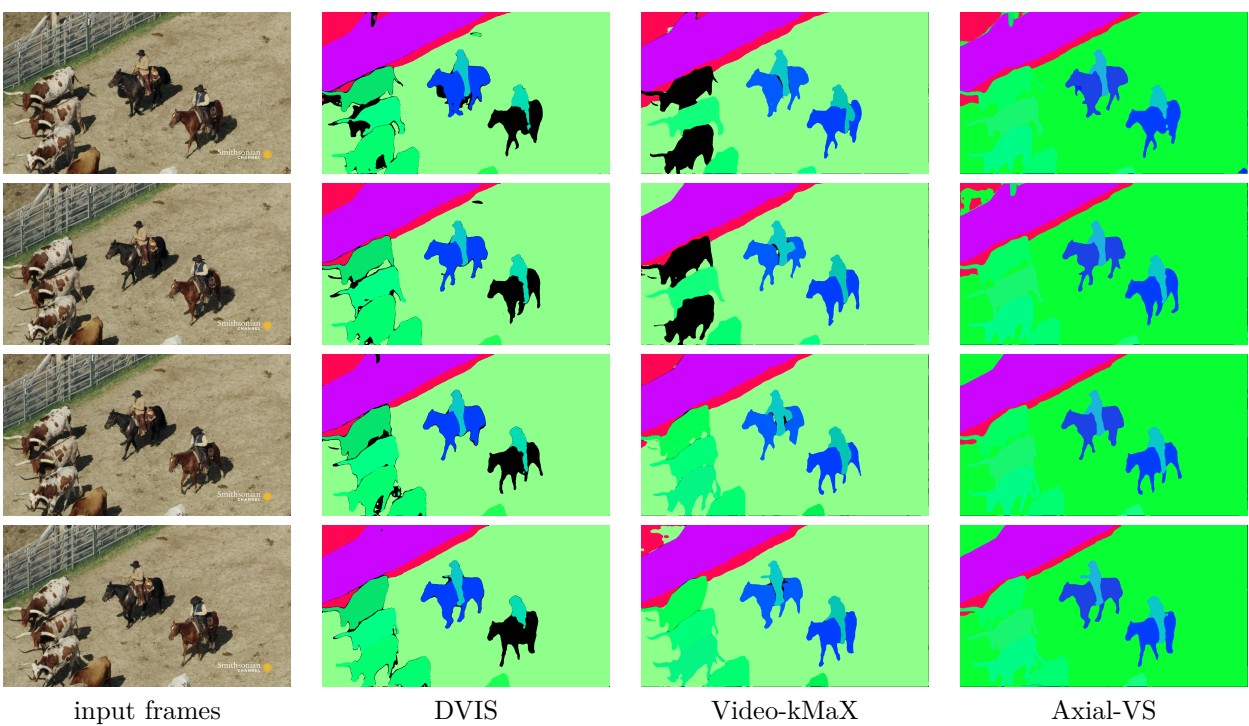

input frames        DVIS        Video-kMaX        Axial-VS

Figure 8: **Qualitative comparisons on videos with unusual viewpoints in VIPSeg.** Axial-VS exhibits consistency in prediction even with an unusual view while DVIS and Video-kMaX fail to consistently detect all animals over time.

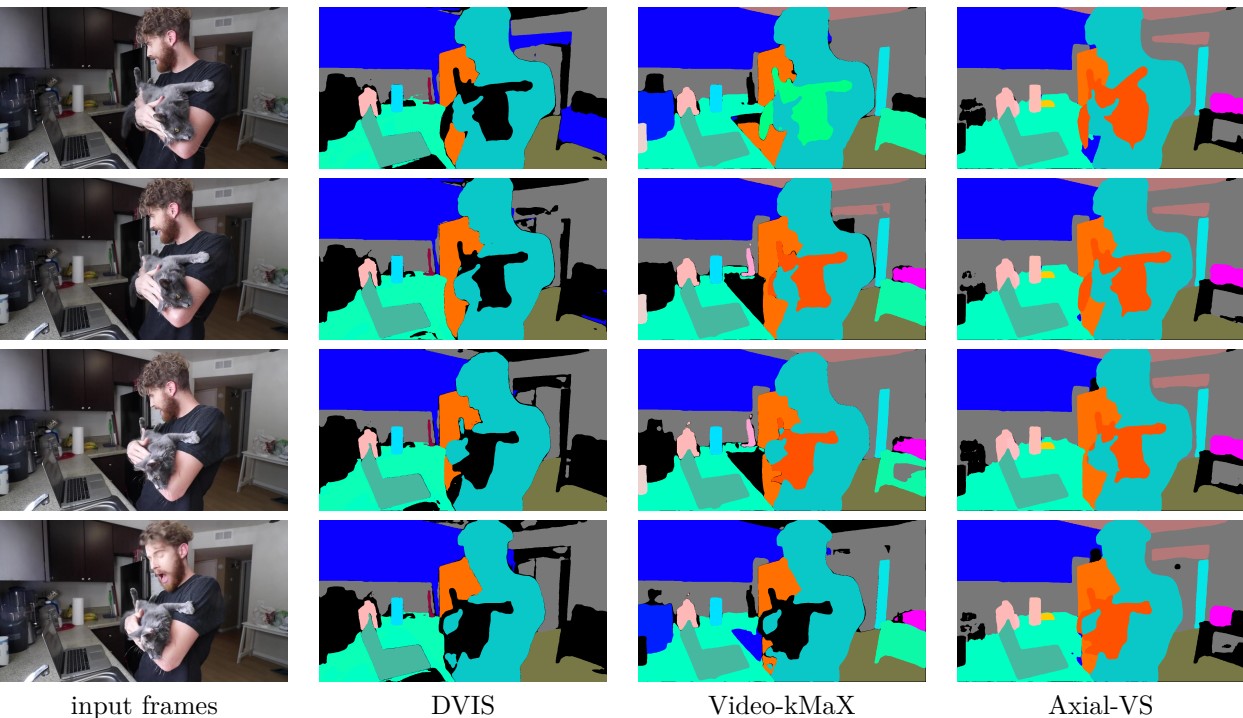

input frames        DVIS        Video-kMaX        Axial-VS

Figure 9: **Qualitative comparisons on videos with complex indoor scenes as background in VIPSeg.** Axial-VS accurately segments out the boundary of cat with correct classes, while DVIS and Video-kMaX fail.

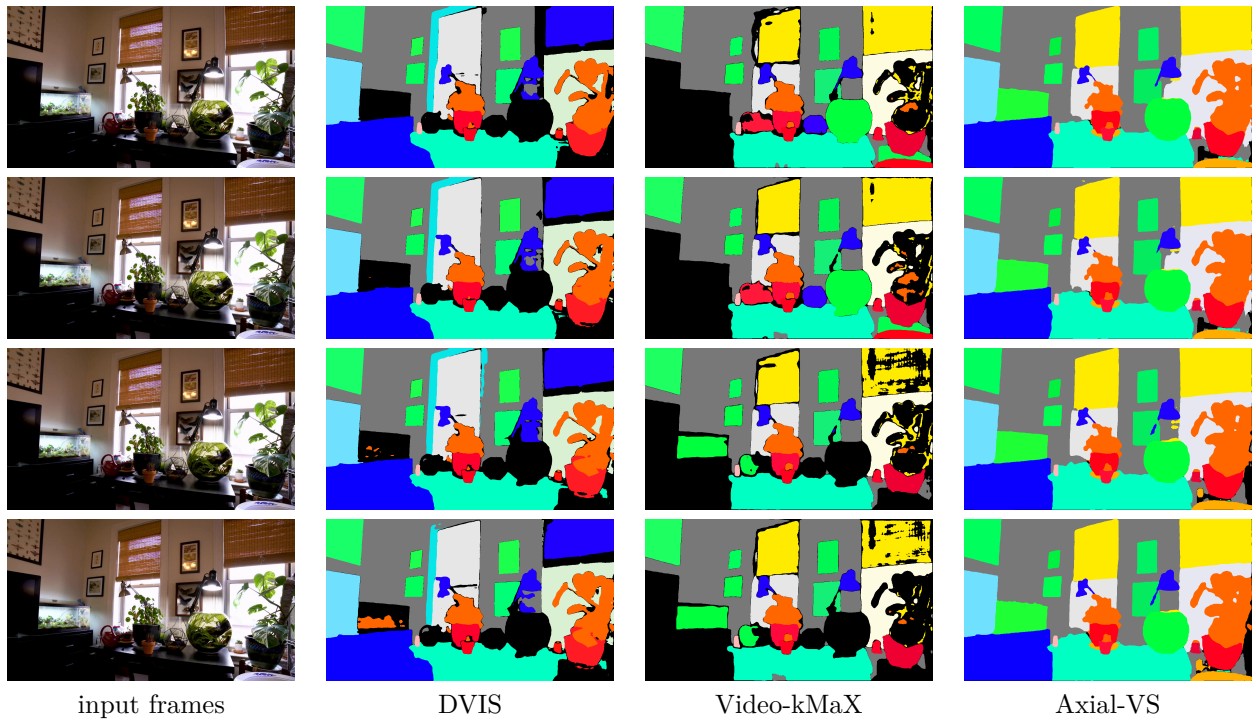

input frames          DVIS          Video-kMaX          Axial-VS

Figure 10: **Qualitative comparisons on videos with light and shade in VIPSeg.** Axial-VS makes accurate and consistent predictions under different illumination situations. DVIS fails at the junction between light and shade (*e.g.*, the fish tank) while Video-kMaX completely fails at dark places.

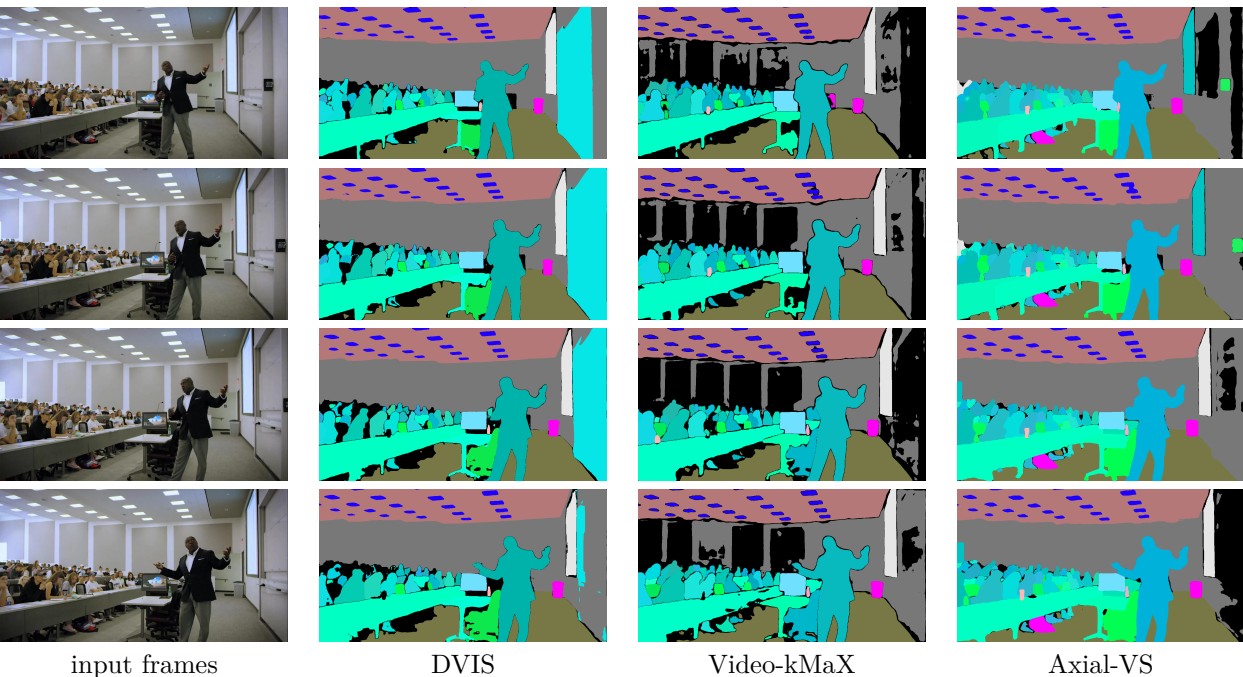

input frames          DVIS          Video-kMaX          Axial-VS

Figure 11: **Qualitative comparisons on videos with multiple instances in VIPSeg.** Axial-VS detects more instances with accurate boundary. DVIS fails to segment out the crowded humans while Video-kMaX performs badly on the stuff classes.

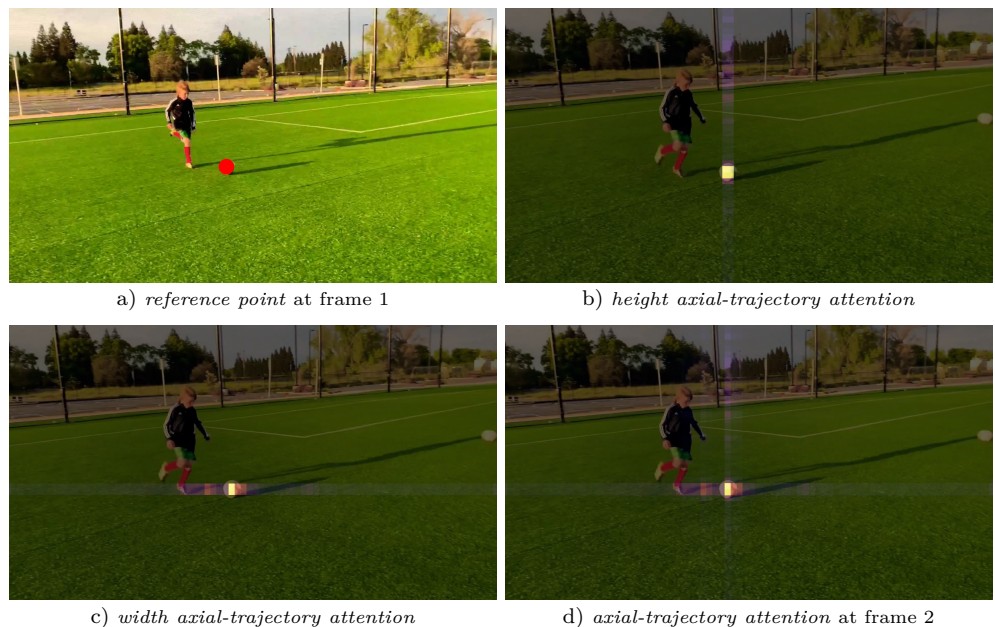

a) *reference point* at frame 1          b) *height axial-trajectory attention*

c) *width axial-trajectory attention*          d) *axial-trajectory attention* at frame 2

Figure 12: **Illustration of Tracking Objects along Axial Trajectories.** In this short clip of two frames depicting the action 'playing football' and the football at frame 1 is selected as the *reference point* (mark in red). We multiply the height and width axial-trajectory attentions to visualize the trajectory of the *reference point* over time.

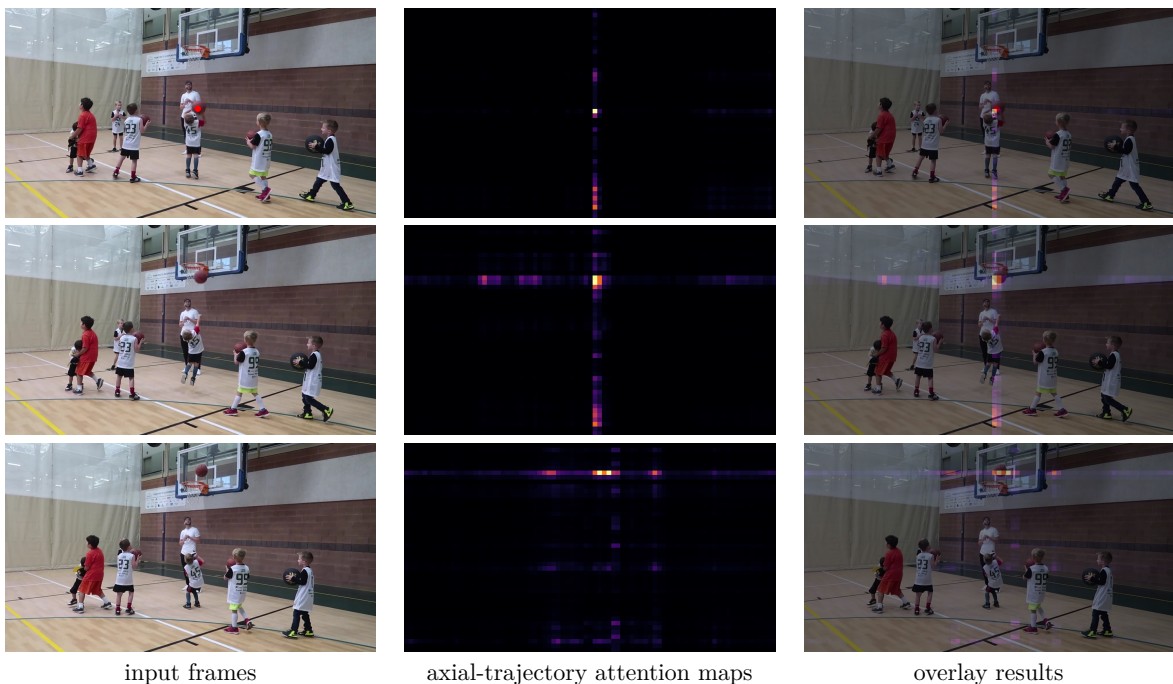

input frames          axial-trajectory attention maps          overlay results

Figure 13: **Visualization of Learned Axial-Trajectory Attention.** In this short clip of three frames depicting the action 'play basketball', the basketball at frame 1 is selected as the *reference point* (mark in red). The axial-trajectory attention can accurately track the moving basketball across frames. Best viewed by zooming in.

third frame and the attention dilutes. Similarly, in Fig. 18, we select the head of the human as the *reference point.* Since the human wears a black jacket with a black hat, the selected reference point has similar but ambiguous appearance to the human body, yielding sparse attention activation in the whole human region.

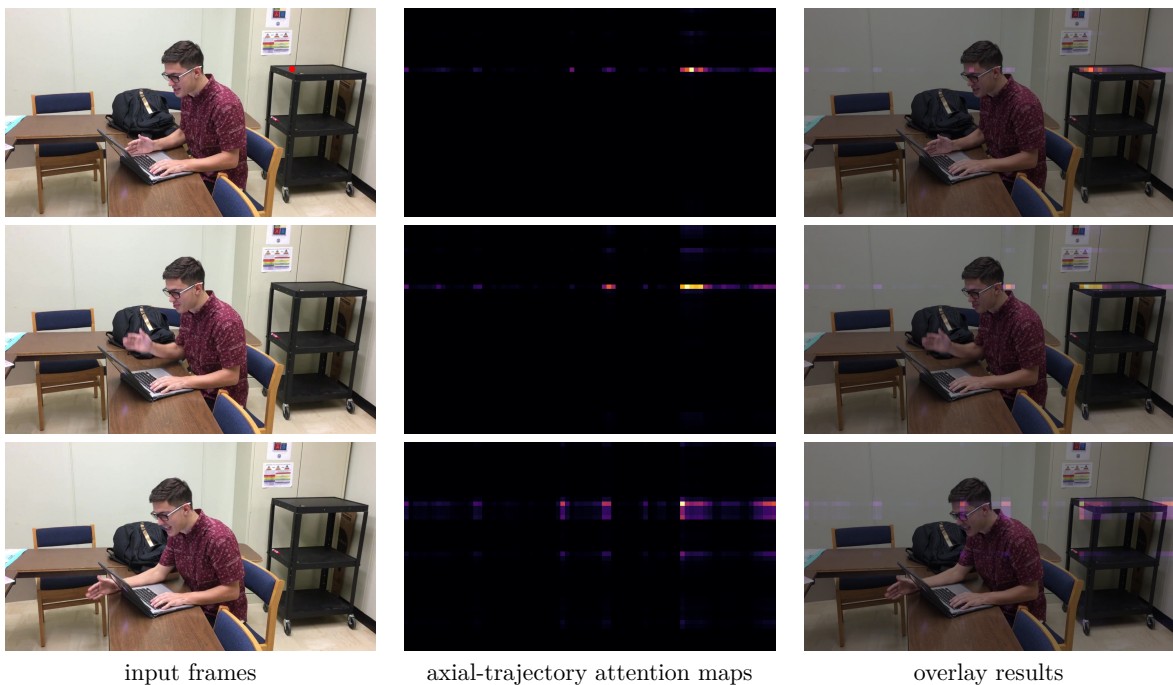

input frames                axial-trajectory attention maps                overlay results

Figure 14: **Visualization of Learned Axial-Trajectory Attention.** In this short clip of three frames depicting a student at class, the right static table at frame 1 is selected as the *reference point* (mark in red). Though the table remains static across the frames, axial-trajectory attention can accurately track it. Best viewed by zooming in.

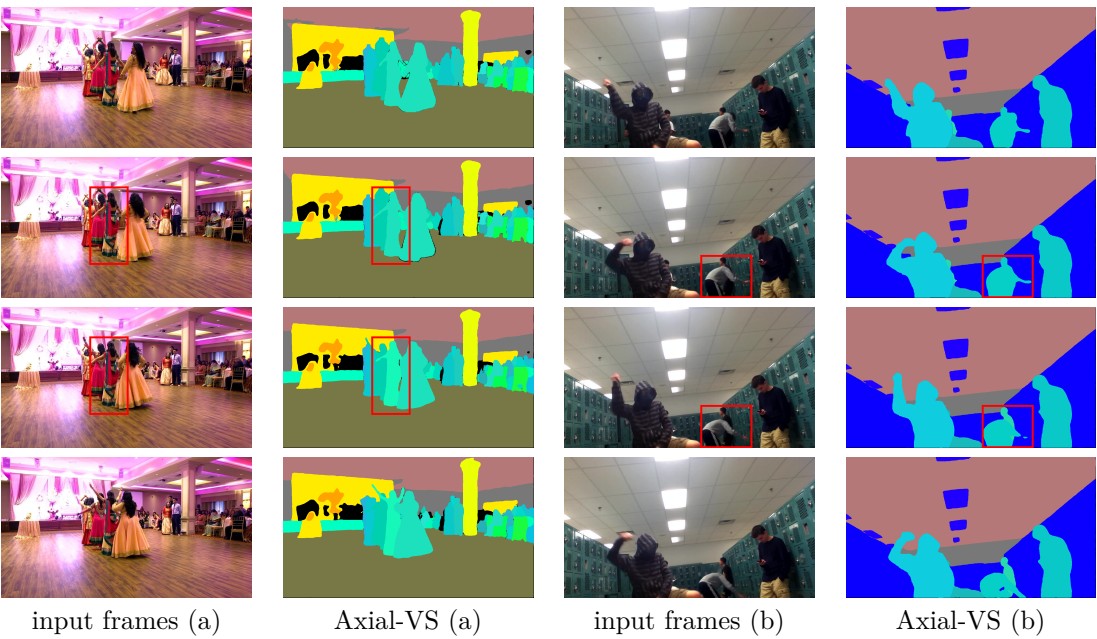

input frames (a)        Axial-VS (a)        input frames (b)        Axial-VS (b)

Figure 15: **Failure modes caused by heavy occlusion.** Axial-VS fails to predict consistent ID for the same instance when there is heavy occlusion. (a) The ID of the human changes between frame 2 and 3; refer to the red box for details. (b) The two humans are recognized as only one until frame 3; refer to the red box for details.. Best viewed by zooming in.

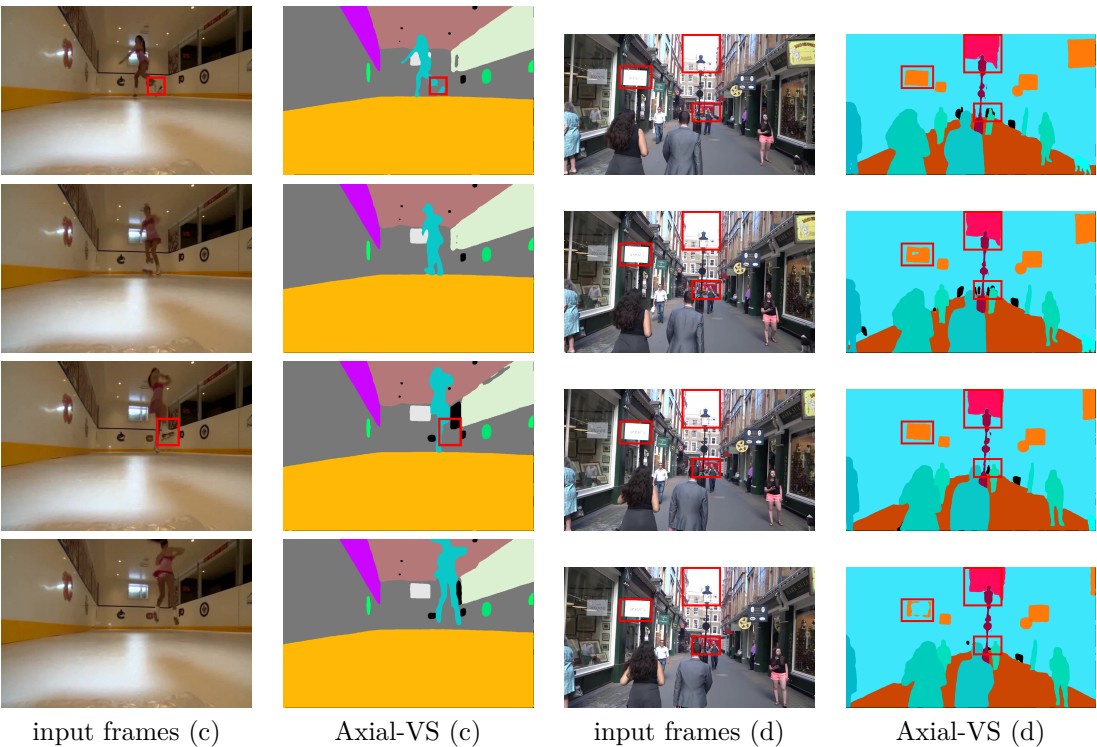

|  |  |  |  |
| :---: | :---: | :---: | :---: |
| input frames (c) | Axial-VS (c) | input frames (d) | Axial-VS (d) |

Figure 16: **Failure modes caused by fast-moving and extreme illumination scenarios.** Axial-VS fails to predict accurate boundary due to the large motion and extreme illumination. (c) The human's leg is not segmented out in frame 1 and 3; refer to the red box for details. (d) The objects under extreme illumination can not be well segmented; refer to the red box for details. Best viewed by zooming in.

## D Limitations

The proposed Axial-VS builds on top of off-the-shelf clip-level segmenters with the proposed within-clip and cross-clip tracking modules. Even though flexible, its performance depends on the underlying employed clip-level segmenter. Additionally, when training the proposed cross-clip tracking module, the clip-level segmenter and the within-clip tracking module are frozen due to insufficient GPU memory capacity, which may lead to a sub-optimal result since ideally, end-to-end training leads to a better performance. We leave it as a future work to efficiently fine-tune the whole model for processing long videos.

## E Datasets

**VIPSeg** (Miao et al., 2022) is a new large-scale video panoptic segmentation dataset, targeting for diverse in-the-wild scenes. The dataset contains 124 semantic classes, consisting of 58 'thing' and 66 'stuff' classes with 3536 videos, where each video spans 3 to 10 seconds. The main adopted evaluation metric is video panoptic quality (VPQ) (Kim et al., 2020) on this benchmark.

**Youtube-VIS** (Yang et al., 2019) is a popular benchmark on video instance segmentation, where only 'thing' classes are segmented and tracked. It contains multiple versions. The YouTube-VIS-2019 (Yang et al., 2019) consists of 40 semantic classes, while the YouTube-VIS-2021 (Yang et al., 2021a) and YouTube-VIS-2022 (Yang et al., 2022) are improved versions with higher number of instances and videos. Youtube-VIS adopts track AP (Yang et al., 2019) for evaluation.

**OVIS** (Qi et al., 2022) is a challenging video instance segmentation dataset with focuses on long videos with 12.77 seconds on average, and objects with severe occlusion and complex motion patterns. The dataset contains 25 semantic classes and also adopt track AP (Yang et al., 2019) for evaluation.

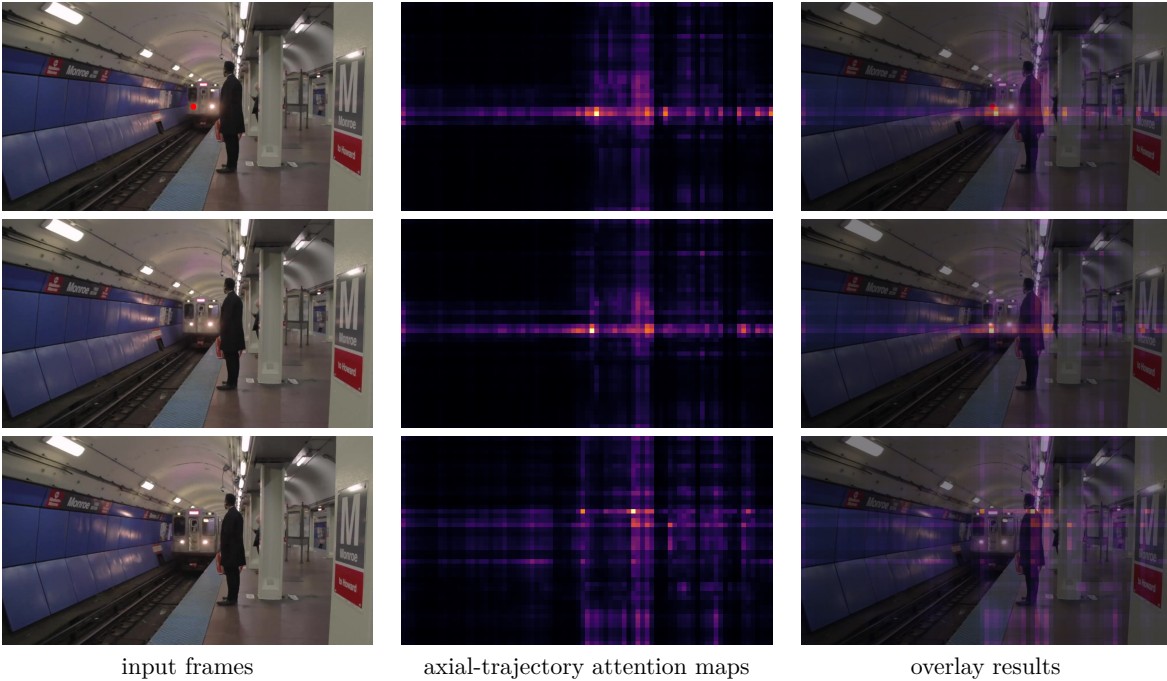


input frames         axial-trajectory attention maps         overlay results


Figure 17: [**Failure mode**] **Visualization of Learned Axial-Trajectory Attention.** In this short clip of three frames depicting a moving subway, the left front light at frame 1 is selected as the *reference point* (mark in red). While the axial-trajectory attention can still more or less capture the same front light at the frame 2, it gradually loses the focus since there are many similar "light" objects in the clip. Best viewed by zooming in.

## F    Broader Impact Statement

This paper introduces Axial-VS, which enhances a standard clip-level segmenter with the proposed axial-trajectory attention, thus advancing the field of video segmentation. While there may be potential societal consequences, however, none of which we feel are necessary to highlight here.

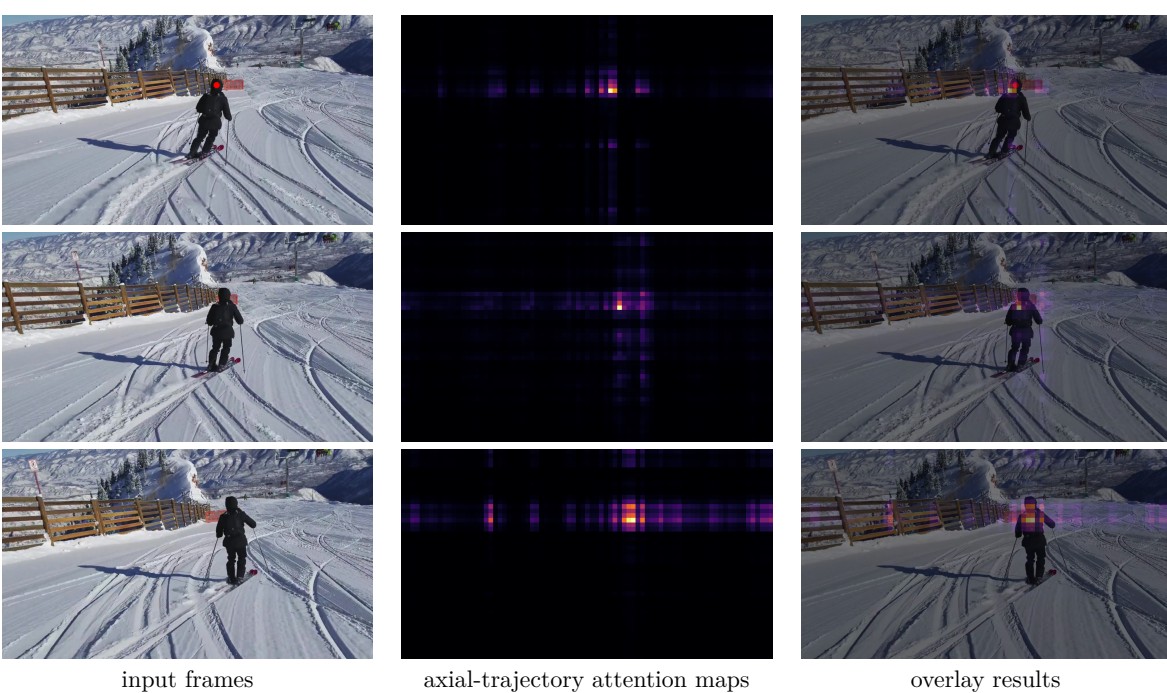

input frames          axial-trajectory attention maps          overlay results

Figure 18: [**Failure mode**] **Visualization of Learned Axial-Trajectory Attention.** In this short clip of three frames depicting the action 'downhill ski', the head of the human at frame 1 is selected as the *reference point* (mark in red). Since the head and the human body have similar appearance, the axial-trajectory attention becomes diluted among the human body. Best viewed by zooming in.

