# OpenReview forum: "A Simple Video Segmenter by Tracking Objects Along Axial Trajectories"
_TMLR — Accepted by TMLR_

### Review · Reviewer_xVXm · 2024-04-28

**Summary Of Contributions:**

The authors' core contribution is the axial-trajectory attention, an extension of a previous method (Patrick et al. 2021). The axial-trajectory attention is a two-phase attention mechanism that initially operates on the T*H (or W) dimension, producing a feature vector for each point in TH across different times, as I understand it. This idea is interesting, and the duplication of time makes it less intuitive and more intriguing than other types of attention methods.

Using the proposed attention mechanism, the authors enhance features for an off-the-shelf clip segmenter and generate better clip queries than the original clip segmenter. They then propose inputting these clip queries into a cross-clip tracking module, which utilizes a similar attention mechanism and a temporal ASPP to refine the results.

**Audience:**

Yes

**Broader Impact Concerns:**

I do not see any additional broader impact concerns.

**Claims And Evidence:**

Yes

**Requested Changes:**

In addition to the weaknesses and nitpicking mentioned above, it would be beneficial to include a section on the failure cases of the model, not just examples in the appendix. Since different models excel in different scenarios, it would be interesting to know when to use, and more importantly, when not to use your proposed method. Specifically, insights into the limitations of the proposed attention mechanism, which could be applicable to other tasks as well, would be very useful.

**Strengths And Weaknesses:**

Pros:
+ The paper is well written and easy to understand with minimal reading of previous works.
+ The method is novel, interesting, and not intuitive, and it improves the results of existing works.
+ The authors provided good ablations and experimental evaluations.
+ The authors have reimplemented the competitors' results, improving their initial results, and applying more modern networks within the said works. Reporting these new, better results is a very good practice and should be commended.

Cons:
- The usage of axial attention in the second part is not clear, and I did not fully understand how the main idea (the axial part across H and W) is translated there. The attention mechanism is similar in the regard where you use the K clips in a similar fashion to the previous usage of time, but it does not reflect the spatial context of the original attention; clarity could be improved there.

Nitpicking:
* Multiple frames in the appendix have really similar colors for adjacent instance segmentations, making it hard to tell the difference.
* The subsubsection titled "Within-Clip Tracking Module" in 3.2 is mostly about the multi-scale attention, and not the entire module; the * title should reflect this.
* Italic 'reference point' (mid-end page 5) gives the notion that it is defined somewhere (e.g., the axial-attention); it is not, and some context should be added to it.
* In Eqn 3, there is a missing apostrophe in the first term on the RHS.

---

> ### Author Response · Authors · 2024-04-29
> **To Reviewer xVXm**
>
> We thank Reviewer xVXm for the constructive comments, and we address the concerns below. After we have received the reviews from all the reviewers, we will proceed with a comprehensive update of our version, adhering to the author guidelines.
>
> > C1: How axial-trajectory attention is applied to the cross-clip tracking module?
>
> We thank the reviewer for bringing up the question and we are happy to clarify the confusion. In the cross-clip tracking module, we apply the axial-trajectory attention to object queries (1D tensors) along the temporal axis (i.e., across all the K clips), and there is no other axial decomposition needed (since it is directly applied to track the N clip object queries across all K clips). Specifically, for a particular query $n$ within a specific clip $k$, Eq. 5 calculates its probabilistic trajectory queries across clips (the affinity of object queries in other clips that correspond to the query $n$) , while Eq. 6 aggregates information along this trajectory path. To summarize, we compute the axial-trajectory attention once for all clip object queries to capture the temporal connections across the entire video.
>
> > N1: Adjacent instance segmentations share similar colors, making it hard to tell the difference.
>
> We thank the reviewer for the suggestion. To aid visualization, we followed the established practices (and used the same visualization scripts) in video panoptic segmentation literature by using the same color for a specific class and introducing *minor color perturbations* to differentiate between instances. Given the extensive range of classes in video panoptic segmentation datasets (e.g., 124 semantic classes for the VIPSeg dataset), achieving clear visualization in videos with multiple instances can be challenging. That being said, we will open-source both the code and pretrained checkpoints, allowing the community to scrutinize our visualization results.
>
> > N2: The subsubsection titled "Within-Clip Tracking Module" in 3.2 is mostly about the multi-scale attention, and not the entire module; the * title should reflect this.
>
> We thank the reviewer for pointing it out. We will update the subsubsection title to ***Multi-Scale Attention*** to more accurately reflect the content of this paragraph in the revised version.
>
> > N3: Italic 'reference point' (mid-end page 5) gives the notion that it is defined somewhere (e.g., the axial-attention); it is not, and some context should be added to it.
>
> We thank the reviewer for pointing it out. In the original draft, we introduced *`reference point'* in the caption of Figure 1 and visualized it in the figure, which denotes the selected location when illustrating the attention computation process. In the revised version, we will incorporate this definition into the main text for clarity.
>
> > N4: In Eqn 3, there is a missing apostrophe in the first term on the RHS.
>
> We are sorry for the confusion of the text description. The absence of an apostrophe is intentional, as the query represents the *selected* trajectory point at time t, while the keys and values are extended along the trajectory path across time. To avoid confusion, we will consider removing the notation $\widetilde{y}_{tt'h}$ in the text description and revising it to *'Specifically, we linearly project the trajectory points and obtain a new set of query-key-value vectors'*.
>
> > R1: Include a section on the failure cases of the model to know when to use and when not to use.
>
> We thank the reviewer for the suggestion. In the original draft, we have already included two paragraphs titled ***Failure Cases for Prediction*** and ***Failure Cases for Learned Axial-Trajectory Attention*** in the appendix (Page 23) to carefully address potential failure scenarios for Axial-VS and Axial-Trajectory Attention, respectively. To summarize, the overall Axial-VS model might face three common error patterns: heavy occlusion, fast-moving objects, and extreme illumination conditions. Additionally, the proposed Axial-Trajectory Attention may encounter challenges if the selected reference point lacks discriminative features, leading to sparse attention activation due to similarities with other points. For detailed discussions, please refer to these two paragraphs in the appendix page 23.

---

> ### Author Response · Authors · 2024-05-11
> **Summary of the Changes in Revised Manuscript**
>
> We thank Reviewer xVXm for the constructive comments. We have carefully considered all of your comments/suggestions and revised our manuscript accordingly. The following summary outlines the key modifications. Please refer to the updated draft and the previous rebuttal for the full discussions on all points.
>
> > N2: The subsubsection titled "Within-Clip Tracking Module" in 3.2 is mostly about the multi-scale attention, and not the entire module; the title should reflect this.
>
> Please see **Subsubsection "Multi-Scale Attention" on Page 6** for the revised subsubsection title.
>
> > N3: Italic 'reference point' (mid-end page 5) gives the notion that it is defined somewhere (e.g., the axial-attention); it is not, and some context should be added to it
>
> Please see **second paragraph of Subsection "Axial-Trajectory Attention" on Page 5** for the added text to introduce 'reference point'.

---

### Review · Reviewer_NYeD · 2024-05-07

**Summary Of Contributions:**

The paper proposes a framework called Axial VS to learn dense video features in an efficient way to facilitate downstream segmentation tasks. It involves factorizing the attention op to operate along the height and width dimensions one after the other rather than exhaustively in one go, thus resulting in a reduced memory footprint. The paper proposes network blocks for space-time attention within clips, and also across clips where it specifically designed for DETR-based video segmentation architectures that rely on object queries.

The proposed framework is applied to two existing models, namely TubeLink and Video-kMax and is evaluated for Video Panoptic Segmentation (VPS) on VIPSeg, and Video Instance Segmentation (VIS) on YouTube-VIS and OVIS. Several ablation experiments are also reported that evaluate the efficacy of individual contributions.

**Audience:**

Yes

**Broader Impact Concerns:**

The paper proposed a low-level video feature learning mechanism. It is hard to see any serious ethical implications of such a work.

**Claims And Evidence:**

Yes

**Requested Changes:**

The paper should be modified to address the weaknesses, i.e.:

1. Discussion about clip length and why the performance deteriorates for higher clip lengths even though temporal attention is explicitly handled in the proposed framework.

2. Analysis of VRAM usage for the proposed framework against existing baselines. Ideally this should be done for a few different feature map resolutions.

3. Some of the most important implementation details should be part of the main text e.g. the feature map resolution at which Axial-VS is applied, number of GPUs utilized for training, total training time.

**Strengths And Weaknesses:**

**STRENGTHS**

Overall, it is a pretty solid paper that it is really well-written and easy to read. The contributions are also well-motivated and strongly backed by experimental evaluation. The workflow for the proposed network blocks is nicely explained with consistent notation and illustrations.

On the experimental side, the evaluation is thorough with 3 datasets for 2 different tasks. The proposed Axial-VS is applied to two existing architectures. For each, a self-trained baseline is created and the difference in performance from both within-clip and across-clip attentions are shown separately. Ablations are also thorough with all the major design choices covered and averages over multiple runs being reported.

**WEAKNESSES**

1. Given that the paper focuses on video-centric feature learning with a focus on the temporal aspect on video, it is a bit of a letdown to see that the performance peaks at a clip length of 2 (Table 7a) which is the bare minimum increment over a single image.

2. Since the motivation behind axial attention across height and width separately is to be efficient in terms of VRAM, there should be some experimental analysis for this e.g. comparison of the proposed Axial-VS to a vanilla baseline with full-fledged space-time attention, and the other existing video attention mechanisms in Fig. 7. I can see some analysis in terms of GFLops in the supplementary, but IMO VRAM would be more interesting and should be briefly discussed in the main text.

---

> ### Author Response · Authors · 2024-05-08
> **To Reviewer NYeD (1/2)**
>
> We thank Reviewer NYeD for the constructive comments, and we address the concerns below. After we have received the reviews from all the reviewers, we will proceed with a comprehensive update of our version, adhering to the author guidelines.
>
> > W1: The performance peaks at a clip length of 2, which is the bare minimum increment over a single image.
>
> We thank the reviewer for bringing up the question. Our method demonstrates overall enhancement of Video-kMaX across various clip lengths, as outlined in Table 7(b). Notably, we observe that a clip length of 2 yields optimal results on Video Panoptic Segmentation (VPS), primarily influenced by the performance variance of the deployed clip-level segmenters (i.e., Video-kMaX). We provide two main hypotheses:
>
> - In existing video panoptic segmentation datasets such as VIPSeg, objects typically exhibit slow movement. Therefore, neighboring frames tend to contain the most informative data, with minimal additional information gained from including more frames.
>
> - The transformer decoders employed in Video-kMaX may encounter challenges when processing larger feature maps associated with longer clip lengths. As presented in their original paper,  Video-kMaX also adopts clip length 2 in their final settings when training on VIPSeg.
>
> Consequently, our Axial-VS model, operating on the same datasets and adhering to the same transformer decoder design, consistently improves performance but attains the highest efficacy at a clip length of 2.
>
> > W2: VRAM comparisons of the proposed Axial-VS to other existing video attention mechanisms.
>
> We thank the reviewer for the suggestion. We provide the VRAM comparison below and will incorporate it into the main text in the revised version. The numbers are obtained by measuring on an input clip resolution of 2x769x1345 with ResNet-50 as the backbone, resulting in the Res4 feature resolution of 2x49x85 and Res5 feature resolution of 2x25x41 (the temporal attention is applied to both Res4 and Res5 features). As shown in the table, our proposed Axial-Trajectory Attn occupies slightly less VRAM compared to MSDeformAttn and notably less than Joint Space-Time Attn or Divided Space-Time Attn, while consistently outperforming all these counterparts. When combining MSDeformAttn and Axial-Trajectory Attn together, our final model requires slightly more VRAM compared to TarViS (MSDeformAttn + Window Space-Time Attn) but significantly outperforms it.
>
> | attention operations                  | VRAM   | GFlops | VPQ  |
> |---------------------------------------|--------|--------|------|
> | -                                     | 11.99G | 354    | 42.7 |
> | Joint Space-Time Attn                 | 25.97G | 493    | 43.2 |
> | Divided Space-Time Attn               | 19.58G | 430    | 43.6 |
> | MSDeformAttn                          | 14.15G | 432    | 44.5 |
> | Axial-Trajectory Attn                 | 13.81G | 443    | 44.7 |
> | MSDeformAttn + Window Space-Time Attn | 15.15G | 476    | 44.9 |
> | MSDeformAttn + Axial-Trajectory Attn  | 15.59G | 481    | 46.1 |
>
> We additionally report the VRAM comparison on an input clip resolution of 2x513x897 with ResNet-50 as the backbone below, resulting in the Res4 feature resolution of 2x33x57 and Res5 feature resolution of 2x17x29. As observed, the trend aligns with the previous table.
>
> attention operations                  | VRAM   |
> |---------------------------------------|--------|
> | -                                     | 6.74G |
> | Joint Space-Time Attn                 | 9.87G |
> | Divided Space-Time Attn               | 8.58G |
> | MSDeformAttn                          | 7.75G |
> | Axial-Trajectory Attn                 | 7.57G |
> | MSDeformAttn + Window Space-Time Attn | 8.21G |
> | MSDeformAttn + Axial-Trajectory Attn  | 8.38G |

---

> ### Author Response · Authors · 2024-05-08
> **To Reviewer NYeD (2/2)**
>
> > R1: Discussion about clip length and why the performance deteriorates for higher clip lengths even though temporal attention is explicitly handled in the proposed framework.
>
> We thank the reviewer for bringing up the question and please refer to W1 for the detailed discussions. We will incorporate them into the main text in the revised version.
>
> > R2: Analysis of VRAM usage for the proposed framework against existing baselines.
>
> We thank the reviewer for the suggestion and please refer to W2 for the VRAM comparisons on two different input resolutions. We will incorporate them into the main text in the revised version.
>
> > R3: Some of the most important implementation details should be part of the main text.
>
> We thank the reviewer for the suggestion. We provide the training details below and will incorporate them into the main text in the revised version. Additionally, we will open-source the full code, including training configs and pretrained checkpoints, allowing the community to reproduce our results.
>
> * For the VPS task with ResNet-50 as the backbone, the training protocol follows Video-kMaX. Specifically, we train our near-online Axial-VS on VIPSeg dataset with a clip size of 2x769x1345 and a batch size of 32 using 16 V100 32G GPUs for 40k iterations. The training process takes approximately 13 hours. Additionally, we train our offline Axial-VS on VIPSeg with a video size of 24x769x1345 (12 clips, each consisting of 2 frames) and a batch size of 16 using 8 A100 80G GPUs for 15k iterations. The training process takes approximately 10 hours.
>
> * For the VIS task with ResNet-50 as the backbone, the training protocol follows Tube-Link. Specifically, we train our near-online Axial-VS on Youtube-VIS with a batch size of 8 clips (each clip has 4 frames) using 8 V100 32G GPUs for 15k iterations. We follow the literature to randomly resize the shortest edge of each clip to a given size among [288, 320, 352, 384, 416, 448, 480, 512]. The training process takes approximately 7 hours. Additionally, we train our offline Axial-VS on Youtube-VIS with a batch size of 8 videos (each video has 20 frames, equivalent to 5 clips) using 8 V100 32G GPUs for 10k iterations. The training process takes approximately 4 hours.

---

> ### Author Response · Authors · 2024-05-11
> **Summary of the Changes in Revised Manuscript**
>
> We thank Reviewer NYeD for the constructive comments. We have carefully considered all of your comments/suggestions and revised our manuscript accordingly. The following summary outlines the key modifications. Please refer to the updated draft and the previous rebuttal for the full discussions on all points.
>
> > R1 & W1: Discussion about clip length and why performance deteriorates for higher clip lengths even though temporal attention is explicitly handled in the proposed framework.
>
> Please see **Subsubsection "Clip Length and Clip Sampling Range" on Page 12** for the discussions on the possible reasons why performance deteriorates for higher clip lengths.
>
> > R2 & W2: Analysis of VRAM usage for the proposed framework against existing baselines.
>
> Please see **Subsection B.1 on Page 18 in Appendix** for the VRAM comparisons of the proposed Axial-VS to other existing video attention mechanisms.
>
> > R3: Some of the most important implementation details should be part of the main text.
>
> Please see **Section A on Page 17 in Appendix** for the provided implementation details.

---

> > ### Comment · Reviewer_NYeD · 2024-05-15
> > **Final Comments**
> >
> > Looking at the othe reviews, the overall feedback is generally positive and there seem to be no further major weaknesses that I initially missed. In my initial review, I mainly asked for elaboration on the choice of clip length, quantitative analysis for VRAM usage, and request for some implementation detail. All three points have been satisfiably addressed in the comments above. I am therefore giving this paper a **positive/accept** rating.

---

> > > ### Author Response · Authors · 2024-05-15
> > > **Thanks to Reviewer NYeD**
> > >
> > > We sincerely thank the reviewer for the valuable review and feedback.

---

### Review · Reviewer_Lm39 · 2024-05-09

**Summary Of Contributions:**

This paper presents "Axial-VS," an approach that can enhance video segmentation by tracking objects along axial trajectories in videos. Technically, the authors propose an axial-attention module that performs self-attention along the height- or width-wise attention. The authors also further decompose the task into two main sub-tasks: within-clip segmentation and cross-clip tracking.

Inside the within-clip Segmentation, Axial-VS uses axial-trajectory attention to track objects within individual clips; while in the Cross-Clip Tracking module, the framework applies axial-trajectory attention to clip-level segmenters’ object queries, enabling effective tracking of objects across different clips. This helps achieve consistent segmentation throughout the video.

The proposed method can improve the baseline models with less memory and time consumption, however, there is lack of evidence supporting this claim.

**Audience:**

Yes

**Broader Impact Concerns:**

The authors briefly discussed the broader impact. No concerns are found there.

**Claims And Evidence:**

Yes

**Requested Changes:**

The texts should be further polished. Moreover, additional experimental evaluations should be included, see the weaknesses:
- Running time and memory consumption should be reported.
- Results with swin-L or other baseline models with ConvNeXt should be reported for fair comparison for Table 1 (b).
- Missing ablation study: if full attention is used inside the within-clip tracking module, will the performance be improved? How about the running time and memory consumption?

**Strengths And Weaknesses:**

Strengths:
- The video demos look great.
- The code will be released.
- The results, both qualitative and quantitative ones are promising.

Weaknesses:
- the writing of the paper should be further polished: E.g. "GPU Out-Of-Memory errors" in the abstract are informal; Missing words in the first sentence of Paragraph 3 (intro).
- In the last second paragraph of the introduction, why does the axial attention "concurrently consider spatial and temporal information"? The answer can be found in the method section but it's confusing in the introduction part.
- The metric VPQ is never defined in the main paper.
- Running time and memory consumption should be reported.

Other concerns:
- As claimed by the authors, the decomposition of the task and the axial attention modules help to address the memory consumption. However, some sparse attention modules and memory-efficient attention operations can help to reduce the computational cost but maintain the performance of the full attention operations. How do the proposed axial-attention modules compare with these methods?
- The object queries remain the same for different video clips. Given that a transformer decoder is used inside the framework to produce the output queries, will using the output object query from clip t-1 as the input object query for clip t help improve the performance? i.e. if we update the object queries for different video clips, will the performance be further improved?
- Missing ablation study: if full attention is used inside the within-clip tracking module, will the performance be improved? How about the running time and memory consumption?
- Results with swin-L or other baseline models with ConvNeXt should be reported for fair comparison for Table 1 (b).

**Overall, I think this paper should be further revised.**

---

> ### Author Response · Authors · 2024-05-11
> **To Reviewer Lm39 (1/3)**
>
> We thank Reviewer Lm39 for the constructive comments, and we carefully address the concerns below.
>
> > W1: The writing of the paper should be further polished: E.g. "GPU Out-Of-Memory errors" in the abstract are informal; missing words in the first sentence of Paragraph 3 (intro).
>
> We respectfully disagree with the reviewer. Both the other two reviewers consider that our paper is well-written (please refer to the strengths in their reviews). We would like to seek more concrete comments from the reviewer. In particular,
>
> 1. Why are "GPU Out-Of-Memory errors" informal? Can the reviewer please suggest the formal way to express OOM (out-of-memory)?
>
> 2. What words are missing in the first sentence of Paragraph 3 (intro)? We do not think there are any serious grammatical errors in that sentence that require ***further polished***.
>
> Finally, we kindly request the reviewer to provide more concrete comments regarding what are the remaining writing issues in the draft.
>
> > W2: In the last second paragraph of the introduction, why does the axial attention "concurrently consider spatial and temporal information"?
>
> We would like to kindly correct the reviewer that it is "axial trajectory attention" instead of "axial attention." In the sentence that is right before *"concurrently considering spatial and temporal information"*, we have clearly mentioned that *"The axial-trajectory attention is designed to learn the temporal correspondences between neighboring frames by estimating the motion paths sequentially along the height- and width-axes."* For your reference, the "temporal correspondences between neighboring frames" and "estimating the motion paths sequentially along the height- and width-axes" correspond to "temporal" and "spatial" information, respectively.
>
> > W3: The metric VPQ is never defined in the main paper.
>
> We thank the reviewer for pointing it out. We first emphasize that we do not propose any new metric for all the experiments, but instead adopt the standard metrics.
>
> For reference, the metric video panoptic quality (VPQ) is proposed and defined in the paper "Video Panoptic Segmentation" [1] to measure the quality of video panoptic segmentation models. It extends the idea of panoptic quality (PQ) metric in image panoptic segmentation [2].
>
> [1] Dahun Kim, et al., Video Panoptic Segmentation, CVPR 2020
>
> [2] Alexander Kirillov, et al., Panoptic Segmentation, CVPR 2019
>
> > W4: Running time and memory consumption should be reported.
>
> We thank the reviewer for the suggestion. We provide the running time (FPS) and memory consumption (VRAM) comparisons below and incorporate it into the main text in the revised version. The numbers are obtained by measuring on an input clip resolution of 2x769x1345 with ResNet50 as the backbone using an A100 GPU, resulting in the Res4 feature resolution of 2x49x85 and Res5 feature resolution of 2x25x41 (the temporal attention is applied to both Res4 and Res5 features). As shown in the table, we observe:
>
> * On the running time (FPS) side, our proposed Axial-Trajectory Attn runs notably quicker than Joint Space-Time Attn and slightly slower compared to MSDeformAttn and Divided Space-Time Attn, while consistently outperforming all these counterparts in VPQ. When combining MSDeformAttn and Axial-Trajectory Attn together, our final model runs roughly the same speed compared to TarViS (MSDeformAttn + Window Space-Time Attn) but significantly outperforms it in VPQ.
>
> * On the memory consumption (VRAM) side, our proposed Axial-Trajectory Attn occupies slightly less VRAM compared to MSDeformAttn and notably less than Joint Space-Time Attn or Divided Space-Time Attn, while consistently outperforming all these counterparts in VPQ. When combining MSDeformAttn and Axial-Trajectory Attn together, our final model requires slightly more VRAM compared to TarViS (MSDeformAttn + Window Space-Time Attn) but significantly outperforms it in VPQ.
>
> | attention operations                  | VRAM   | FPS  | GFlops | VPQ  |
> |---------------------------------------|--------|------|--------|------|
> | -                                     | 11.99G | 14.3 | 354    | 42.7 |
> | Joint Space-Time Attn                 | 25.97G | 10.3 | 493    | 43.2 |
> | Divided Space-Time Attn               | 19.58G | 12.6 | 430    | 43.6 |
> | MSDeformAttn                          | 14.15G | 12.5 | 432    | 44.5 |
> | Axial-Trajectory Attn                 | 13.81G | 11.7 | 443    | 44.7 |
> | MSDeformAttn + Window Space-Time Attn | 15.15G | 10.5 | 476    | 44.9 |
> | MSDeformAttn + Axial-Trajectory Attn  | 15.59G | 10.5 | 481    | 46.1 |

---

> ### Author Response · Authors · 2024-05-11
> **To Reviewer Lm39 (2/3)**
>
> > C1: How do the proposed axial-attention modules compare with other sparse attention modules and memory-efficient attention operations?
>
> We thank the reviewer for the question. Firstly, we would like to clarify that our primary objective in this work is not solely focused on designing an efficient attention mechanism. Our core aim is to enhance the tracking capability of ***clip-level segmenters***, encompassing improvements at both ***within-clip*** and ***cross-clip*** levels. The introduction of axial-trajectory attention represents a significant exploration in advancing this specific pursuit. As discussed in the Related Work section, while there have been numerous explorations into designing efficient attention mechanisms for ***video classification***, the attention design for ***video segmentation*** has seen relatively fewer studies. Previous works such as VITA and DVIS primarily focus on attention mechanisms related to object queries. In contrast, when considering dense feature maps, the most relevant work to ours is TarVIS, which introduces a temporal neck by combining MSDeformAttn and Temporal Window Attention. However, as shown in Table 4(a), TarVIS performs worse than the proposed axial-trajectory attention despite having roughly the same computational cost. We hypothesize that this discrepancy arises because, as a dense prediction task, video segmentation inherently benefits from trajectory attention due to its ability to capture object trajectories over time.
>
> > C2: If we update the object queries for different video clips, will the performance be further improved?
>
> We thank the reviewer for the question. Recent study [3] in the field has revealed that for frame-level video segmenters, utilizing the instance representation from the previous frame as the initial representation for the current frame can introduce ambiguous information, thereby making segmentation more challenging. It is hypothesized that this challenge arises primarily due to the high similarity in instance representations between adjacent frames, despite differences in position, shape, size, etc. Similarly, using the output object query from previous clip $k-1$ as the input object query for current clip $k$ may encounter similar challenges, potentially requiring additional exploration to resolve ambiguities. This aspect serves as an intriguing direction for future exploration in the field.
>
> [3] Tao Zhang, et al., DVIS: Decoupled Video Instance Segmentation Framework, ICCV 2023
>
> > C3: If full attention is used inside the within-clip tracking module, will the performance be improved? How about running time and memory consumption?
>
> We thank the reviewer for the question. In the original draft, we already presented the experimental results of utilizing full attention (i.e., Joint Space-Time Attn) in Table 4(a). As demonstrated in the table, employing full attention marginally enhances the performance of the baseline Video-kMaX from 42.7 VPQ to 43.2 VPQ. Please refer to W4 for the running time and memory consumption comparison with other attention mechanisms, where full attention occupies significantly larger GPU memory and runs significantly slower.
>
> > C4: Results with swin-L or other baseline models with ConvNeXt should be reported for fair comparison for Table 1 (b).
>
> We thank the reviewer for the suggestion. However, we respectfully disagree on this point. Firstly, we believe that our experiments provide sufficient evidence to support our claims regarding the effectiveness of our proposed within-clip and cross-clip tracking modules, enhanced by axial-trajectory attention. These enhancements have been systematically verified in our experiments, as evidenced by the results presented in Table 1(a), 2(a), 2(b), 2(c), 4(a), and 4(b), which showcase the performance improvements over the solid reproduced baseline clip-level segmenters (Video-kMaX and Tube-Link). Additionally, in Tab 1(b), equipping other video segmenter (e.g., DVIS, ***non clip-level*** segmenter) with other backbones (e.g., ConvNeXt) or our deployed clip-level segmenter baseline Video-kMaX with Swin-L backbone is beyond our claims, as it is non-trivial to simply replace their backbones and demonstrate similar performances. In Tab. 1(b), we simply would like to demonstrate that "where does our model stand in the benchmark, compared to other methods". We believe that the common backbone ResNet50 is sufficient to demonstrate the difference between our methods and state-of-the-art methods.

---

> ### Author Response · Authors · 2024-05-11
> **To Reviewer Lm39 (3/3)**
>
> > R1: Running time and memory consumption should be reported.
>
> We thank the reviewer for the suggestion. Please refer to W4 and the revised version for the detailed running time and memory consumption comparison.
>
> > R2: Results with swin-L or other baseline models with ConvNeXt should be reported for fair comparison for Table 1 (b).
>
> We thank the reviewer for the suggestion. We respectively disagree with the point and please refer to C4 for the detailed discussions.
>
> > R3: Missing ablation study: if full attention is used inside the within-clip tracking module, will the performance be improved? How about running time and memory consumption?
>
> We thank the reviewer for the suggestion. In the original draft, we already provided the experimental results of using full attention (i.e., Joint Space-Time Attn) in Table 4(a). Please refer to W4 and the revised version for the running time and memory consumption comparison.

---

> ### Author Response · Authors · 2024-05-11
> **Summary of the Changes in Revised Manuscript**
>
> We thank Reviewer Lm39 for the constructive comments. We have carefully considered all of your comments/suggestions and revised our manuscript accordingly. The following summary outlines the key modifications. Please refer to the updated draft and the previous rebuttal for the full discussions on all points.
>
> > R1 & W4: Running time and memory consumption should be reported.
>
> Please see **Subsection B.1 on Page 18 in Appendix** for the running time and memory consumption comparisons.
>
> > R3 & C3: Missing ablation study: if full attention is used inside the within-clip tracking module, will the performance be improved? How about running time and memory consumption?
>
> Please see **Subsection B.1 on Page 18 in Appendix** for performance (VPQ), running time and memory consumption of the full attention variant.

---

> ### Comment · Reviewer_Lm39 · 2024-05-18
> **Thanks for the response**
>
> I thank the authors for their response. The response addresses part of my concerns and the additional experimental evaluations look promising. Here are some following discussions based on the authors' response:
>
> W1: A few concrete comments for the introduction, the other sections should also be polished:
> - "GPU Out-Of-Memory errors" is more like an error message from program running instead of from an academic paper. Using ``...leading to insufficient GPU memory capacity'' may be better.
> - Paragraph 2: avoid using "(xx)" unless referring to figures and tables, or for abbreviation, buts using ", xx," instead
> - Paragraph 3: The first sentence did not connect well with the preceding sentence/paragraph: what is the "the whole pipeline"? Why do you need "concretely"?
> - Fig.1 is hard to read: are b,c,d the results with attention masks overlaid on the images? Which axis is used?
>
>
> W3: I understanding VPQ is a standard metric but it should be introduced in the paper. "VPQ (video panoptic
> quality)" is introduced in the very end (page 26).
>
> W4: Thanks for providing these results! I am curious why the axial-trajectory attention even works better than the joint space-time attention? I assume the joint space-time attention should be the most powerful since it takes the information from both axises?
>
> C2: A few other papers in pixel-wise tracking also show that using the instance representation from the previous frame as the initial query lead to strong results, e.g. [A] and their following works. Could the authors do such an ablation and compare the results with different designs?
>
> [A]TrackFormer: Multi-Object Tracking with Transformers: https://arxiv.org/abs/2101.02702
>
> C4: I totally agree that the overall experiments provide sufficient evidence to support the claim of this paper. However, the results in each table should be self-consistent and under a fair comparison setting. I still believe a fair comparison should be reported and it's easy to get a result for the proposed method with swin-L, right?

---

> > ### Author Response · Authors · 2024-05-20
> > **Second Rebuttal to Reviewer Lm39 (1/3)**
> >
> > We thank Reviewer Lm39 for the constructive comments, and we carefully address the concerns below.
> >
> > > W1: A few concrete comments for the introduction, the other sections should also be polished.
> >
> > We thank the reviewer for the additional suggestions that were missing in the initial review, and address each comment below:
> >
> > > W1.1:  "...leading to insufficient GPU memory capacity" may be better.
> >
> > In our humble opinion, both "GPU Out-Of-Memorry errors" and "insufficient GPU memory capacity" refer to the same thing, while the former expression is easier to understand. Additionally, "out of memory" has also been used in multiple published papers [A, B, C, D]. Nevertheless, based on the suggestion, we have replaced "GPU Out-Of-Memory errors" with "insufficient GPU memory capacity" in the **Section "Abstract" on Page 1**, **Table 5 on Page 11** and **Section "Limitations" on Page 26**.
> >
> > [A] Chen Meng, et al., Training Deeper Models by GPU Memory Optimization on TensorFlow, In Proc. of ML Systems Workshop in NeurIPS 2017
> >
> > [B] Yang You, et al., ImageNet Training in Minutes, ICPP 2018
> >
> > [C] Justin Cui, et al., Scaling Up Dataset Distillation to ImageNet-1K with Constant Memory, ICML 2023
> >
> > [D] Yidong Wang, et al., Exploring Vision-Language Models for Imbalanced Learning, IJCV 2023
> >
> > > W1.2: avoid using "(xx)" unless referring to figures and tables, or for abbreviation, buts using ", xx," instead
> >
> > It is worth noting that the writing style of including words in the parentheses (not just limited to figures, tables, or abbreviations) is ***commonly seen*** in published papers [E, F, G, H], which is used as a way to provide more explanations for the context. As a result, we respectfully disagree with the reviewer that we need to replace all our "(xx)" with ", xx", which significantly makes our draft hard to read. We would like to seek the second opinion from the ***Action Editor*** to confirm whether or not it is improper to use the format of "(xx)" in the writing.
> >
> > [E] Kaiming He, et al., Deep Residual Learning for Image Recognition, CVPR 2016
> >
> > [F] Kaimin He, et al., Momentum Contrast for Unsupervised Visual Representation Learning, CVPR 2020
> >
> > [G] Alexey Dosovitskiy, et al., An Image is Worth 16x16 Words: Transformers for Image Recognition at Scale, ICLR 2021
> >
> > [H] Zhuang Liu, et al., A ConvNet for the 2020s, CVPR 2022
> >
> > > W1.3: What is the "the whole pipeline"? Why do you need "concretely"?
> >
> > Similar to our response to your W2 in previous rebuttal, by carefully reading the context of written sentences, one should be able to find the solution, concerning "which subject" in the context is referred to by "the". Again, we would like to emphasize that ***both the other two*** reviewers consider that our paper is ***well-written and easy-to-read*** (please refer to the strengths in their reviews). We kindly encourage the reviewer to more carefully read the context in the paragraphs before raising another question regarding the writing.
> >
> > In response to this comment, for your reference, "the whole pipeline" refers to clip-level segmenter framework, which was introduced ***right before*** that paragraph, and "Concretely" refers to our attempt to explain the framework in a concrete way. We think it is quite clear from the context. However, we are happy to replace "the whole pipeline" with "the workflow of clip-level segmenters" in the revised version.
> >
> > > W1.4: Fig.1 is hard to read
> >
> > In order to illustrate both the height- and width-axial trajectory attention in one figure, they are multiplied for visualization and thus the high values (brighter points) correspond to both high attention values in the both height- and width-axis. We have added texts for clarification as "We multiply the learned height and width axial-trajectory attentions and overlay them on frame 2, 3 and 4 to visualize the trajectory of the reference point over time." in the **caption of Fig. 1 on Page 2**. Additionally, we provide the visualization of height- and width-axial-trajectory attention in the new Fig. 12. Finally, larger figures that visualize the learned axial-trajectory attentions can be found in Fig. 13, Fig. 14, Fig. 17, and Fig. 18 in the appendix.
> >
> > Please review the revised version for the changes mentioned above.

---

> > ### Author Response · Authors · 2024-05-20
> > **Second Rebuttal to Reviewer Lm39 (2/3)**
> >
> > > W3: VPQ should be introduced in the paper.
> >
> > We thank the reviewer for the suggestion. We have added "We utilize Video Panoptic Quality (VPQ), as defined in VPSNet [1], and Average Precision (AP), as defined in MaskTrack R-CNN [2], for evaluating the models on VPS and VIS, respectively." in the 1st paragraph in **Section "Experimental Results" on Page 8**. Please see the revised version.
> >
> > [1] Dahun Kim, et al., Video Panoptic Segmentation, CVPR 2020
> >
> > [2] Linjie Yang, et al. Video Instance Segmentation, ICCV 2019
> >
> > > W4: Why does the axial-trajectory attention even work better than the joint space-time attention?
> >
> > We thank the reviewer for raising this question. We hypothesize that the issue arises mainly because performing attention on very dense high-resolution feature maps results in poor attention quality, with a lack of focus on the most informative regions. This is precisely what axial-trajectory attention aims to capture and track. Numerous studies in video classification [3, 4, 5] demonstrate that various types of sparse attention, which attend to fewer regions, can achieve better results than joint space-time attention. We observed the same trend in our ablation studies for video segmentation in Tab. 4a.
> >
> > [3] Gedas Bertasius, et al., Is Space-Time Attention All You Need for Video Understanding?, ICML 2021
> >
> > [4] Mandela Patrick, et al., Keeping Your Eye on the Ball: Trajectory Attention in Video Transformers, NeurIPS 2021
> >
> > [5] Jue Wang and Lorenzo Torresani, Deformable Video Transformer, CVPR 2022
> >
> > > C2: A few other papers in pixel-wise tracking also show that using the instance representation from the previous frame as the initial query leads to strong results. Could the authors do such an ablation and compare the results with different designs?
> >
> > We thank the reviewer for the suggestion and the reference that was missing in the initial review. We would like to emphasize that although it may have the potential to further boost performance, this ablation study is not related to our claims and beyond the scope of our current work, which focuses on ***augmenting off-the-shelf clip-level video segmenters with the proposed axial-trajectory attention at both within-clip and cross-clip levels***. Additionally, it is noteworthy that it is non-trivial to improve performance by naively initializing the queries from the previous frame, which might introduce ambiguous information, as discussed by existing works:
> >
> > 1. In the provided TrackFormer paper, two hyper-parameters, $\sigma_{track}$ and $\sigma_{NMS}$, are required to determine when a detected query in the previous frame can be removed in the new frame. Additionally, they introduce an attention-based short-term re-identification module with a hyper-parameter $\sigma_{track-reid}$ to reactivate queries if they are considered to be inactive and do not contribute to the trajectory until a classification score is higher than $\sigma_{track-reid}$ (which triggers a re-identification).
> >
> > 2. In DVIS [6], the authors also introduce referring cross-attention mechanism to mitigate ambiguous information between adjacent frames when using the instance representation from the previous frame as the initial queries for the next frame.
> >
> > These explorations show that initializing the queries from the previous frame might introduce ambiguous information, which calls for careful design and future research to alleviate these ambiguities.
> >
> > [6] Tao Zhang, et al., DVIS: Decoupled Video Instance Segmentation Framework, ICCV 2023

---

> > ### Author Response · Authors · 2024-05-20
> > **Second Rebuttal to Reviewer Lm39 (3/3)**
> >
> > > C4: I totally agree that the overall experiments provide sufficient evidence to support the claim of this paper. However, a fair comparison should be reported and it's easy to get a result for the proposed method with Swin-L.
> >
> > We thank the reviewer for ***acknowledging that the provided experiments are sufficient to support the claim made in the submission***. We also thank the reviewer for the suggestion. However, we respectfully disagree with the reviewer that ***it's easy*** to get a result with Swin-L or any other backbone that is not supported by the deployed off-the-shelf clip-level segmenter.
> >
> > Specifically, to add the support of Swin-L for Video-kMaX, it would require the following changes:
> >
> > 1. ***Pretrain the image segmenter kMaX-DeepLab [7] with Swin-L on COCO***: Following prior works [6, 8, 9, 10, 11], and discussed in our Sec. A in the appendix, modern video segmenters require an image segmenter pre-trained on COCO as weight initialization. However, the original kMaX-DeepLab does not support Swin-L in their codebase [12, 13]. It is thus challenging and would require significant efforts to replace ConvNeXt-L in kMaX-DeepLab with Swin-L and achieve similar or better performance on COCO image panoptic segmentation.
> >
> > 2. ***Improve Video-kMaX with Swin-L based on the pre-trained kMaX-DeepLab checkpoint in step 1***: For a fair comparison, it then requires us to rebuild the baseline Video-kMaX with Swin-L, utilizing the new kMaX-DeepLab checkpoint obtained in step 1, to achieve similar performance compared to the original version that uses ConvNeXt-L. Unfortunately, this is also not discussed in the Video-kMaX paper and would require significant efforts to verify its effectiveness.
> >
> > As a result, in all our experiments, we simply focus on augmenting the ***off-the-shelf*** clip-level video segmenter with the proposed axial-trajectory attention at both within-clip and cross-clip levels, sticking to the backbone supported by the clip-level video segmenter. It is beyond the scope and our claims to add any other backbone support for the deployed clip-level video segmenters.
> >
> > Nevertheless, as presented in the previous rebuttal, we believe that the common backbone ResNet50 is sufficient to demonstrate the differences between our methods and state-of-the-art methods. However, if the reviewer insists on seeing "Axial-VS results with Swin-L on VIPSeg," we are happy to remove those Swin-L and ConvNeXt results from the table in the main paper.
> >
> > [7] Qihang Yu, et al., kMaX-DeepLab: k-Means Mask Transformer, ECCV 2022
> >
> > [8] De-An Huang, et al., MinVIS: A Minimal Video Instance Segmentation Framework without Video-based Training, NeurIPS 2022
> >
> > [9] Ali Athar, et al., TarViS: A Unified Approach for Target-based Video Segmentation, CVPR 2023
> >
> > [10] Xiangtai Li, et al., Tube-Link: A Flexible Cross Tube Framework for Universal Video Segmentation, ICCV 2023
> >
> > [11] Inkyu Shin, et al., Video-kMaX: A Simple Unified Approach for Online and Near-Online Video Panoptic Segmentation, WACV 2024
> >
> > [12] https://github.com/google-research/deeplab2/blob/main/g3doc/projects/kmax_deeplab.md
> >
> > [13] https://github.com/bytedance/kmax-deeplab

---

### Decision · Action_Editor_wWhx · 2024-06-03

**Recommendation:** Accept as is

**Comment:**

All the reviewers were positive about the paper. They cite extensive experimental evaluation as the main strength of the work as well as the novel design of the proposed axial trajectory attention. The authors engaged with the reviewers and successfully addressed all the concerns that were raised. Thus, I recommender the paper for acceptance to TMLR without any further revisions.

**Audience:**

Yes.

**Claims And Evidence:**

Yes, according to the submitted reviews, the main claims of this paper are supported by convincing experiments.